# Learning Interpretable Options by Identifying Reward Diffusion Bottlenecks in Reinforcement Learning

## Abstract

Bottleneck states, which connect distinct regions of the state space, provide a principled and interpretable basis for constructing temporal abstractions in Hierarchical Reinforcement Learning (HRL). However, existing bottleneck identification methods primarily rely on topological analysis of the state-transition graph, limiting their scalability to high-dimensional or continuous domains. To address this challenge, we introduce Value Power Strength (VPS), a value function-based metric inspired by the analogy between the Bellman equation and Kirchhoff's current law, to quantify bottleneck property via the diffusion of reward in Markov Decision Processes (MDPs). VPS is estimated efficiently using value functions learned from random reward signals and captures reward diffusion bottlenecks in both discrete and continuous state spaces. Leveraging VPS, we design options that guide agents toward or away from bottleneck regions. Experimental results on classic tabular domains, visual GridWorld, and Atari 2600 games demonstrate that the VPS-based framework discovers semantically meaningful subgoals and substantially improves exploration efficiency.

## 1 Introduction

Reinforcement Learning (RL) has achieved remarkable advances across a range of applications. However, it still faces major challenges when rewards are sparse and tasks extend over long horizons (Sutton & Barto, 1998; Ecoffet et al., 2021). Hierarchical Reinforcement Learning (HRL) alleviates these difficulties by decomposing tasks into subtasks through action abstractions or subgoals, thereby shrinking the exploration space and improving sample efficiency (Pateria et al., 2021; Nachum et al., 2019). Empirically, hierarchical agents have delivered marked gains in exploration efficiency and final performance across diverse navigation, manipulation, and visual-control benchmarks (Nachum et al., 2018; Park et al., 2023; Eysenbach et al., 2018; Bagaria et al., 2023).

Within HRL, the option framework casts a temporally extended behavior as a policy equipped with initiation and termination conditions, allowing high-level planning in multi-step units (Sutton et al., 1999). As the core step of the framework, option discovery algorithms are commonly grouped into two classes. In structure-based approaches, the Markov Decision Process (MDP) is embedded in a graph or latent space and subgoals are selected from topological cues: early work exploits cut and centrality metrics to locate bridge states (Menache et al., 2002; Şimşek et al., 2005; Newman, 2005; Şimşek & Barto, 2008), later studies employ spectral analysis of the graph Laplacian to derive options (Machado et al., 2017a;b; Wang et al., 2021; Klissarov & Machado, 2023), and recent variants refine the hierarchy via modularity maximization or latent-space clustering (Evans & Şimşek, 2023; Ramesh et al., 2019). In contrast, objective-optimization approaches formulate option discovery as an explicit optimization problem, such as minimizing planning or cover time, and furnish theoretical performance guarantees through mixed-integer programming or provable approximations (Solway et al., 2014; Jinnai et al., 2019a;b; Ivanov et al., 2025). While the objective-optimization methods offer formal bounds on efficiency, structure-based methods typically yield subgoals that align with salient environmental landmarks, providing clearer semantics and stronger interpretability.

Bottleneck states, i.e. bridges that connect weakly coupled regions of the state space, provide clear semantics and thus form a natural intrinsic target for option design. Prevailing identification methods

cast the MDP as a graph and infer bottlenecks via topological criteria: betweenness centrality and its efficient computation (Freeman, 1977; Newman, 2005; Brandes, 2001), cut-based measures such as Q-cut and normalized cuts (Şimşek et al., 2005; Menache et al., 2002; Shi & Malik, 2000). Although effective on tabular domains, these graph-centric pipelines do not scale gracefully to high-dimensional or continuous state spaces, and the resulting centrality or cut scores must be computed outside the value function-learning loop, making them difficult to integrate seamlessly into standard RL paradigm.

Recent work has shown that value functions in MDP can be interpreted as discrete potential fields, with the Bellman equation taking the form of a Poisson-type equation on the state-transition graph and closely relating to graph Laplacians and random walks (Mahadevan & Maggioni, 2007; Machado et al., 2017a; Wang et al., 2021). At the same time, classical circuit theory provides a concrete physical realization of such potential fields: Kirchhoff's laws and Ohm's law define currents, voltages, and power dissipation on electrical networks, which can also be viewed as weighted graphs (Doyle & Snell, 1984; Chung, 1997; Chandra et al., 1989; Tetali, 1991). This parallel suggests that several key physical quantities in electrical networks admit meaningful counterparts in RL, even if they have received little attention so far.

To derive a bottleneck indicator that is fully compatible with the RL optimization loop and does not rely on an explicit state-transition graph, we exploit the structural correspondence between the Bellman equation and Kirchhoff's current law, i.e., the total incoming current at a node equals the total outgoing current, revealing how local flows and energy dissipation concentrate in a network. Option discovery fundamentally depends on the choice of variables used to define task hierarchies, and variables that are central in circuit theory yet rarely examined in RL may carry significant potential as new foundations for hierarchical structure. By mapping node power in resistive networks to the Bellman update, we propose Value Power Strength (VPS), a value-based measure that captures bottlenecks of reward diffusion in the state space. To construct VPS-based options, we inject multiple independent noise reward functions, estimate the corresponding value functions and VPS scores, and use VPS as an intrinsic potential guiding policies toward or away from high-VPS states. This process relies only on standard value estimation, requires no explicit graph construction, and naturally extends to high-dimensional settings through function approximation. Experiments show that the framework identifies meaningful subgoals and enhances exploratory behavior.

The main contributions of this paper include: (1) a value function–based metric, VPS, is proposed to quantify state-level bottlenecks without recourse to explicit graph analysis; (2) estimation procedures applicable to both discrete and continuous state spaces are derived and, under mild assumptions, are shown to converge almost surely to the true VPS; and (3) a VPS-based option discovery scheme is designed, yielding semantically meaningful, interpretable temporally extended actions that enhance the exploration efficiency.

## 2 PRELIMINARIES

RL focuses on training agents to make sequential decisions by interacting with an environment. The process is commonly modeled as an MDP (Puterman, 1994), defined by a tuple $(\mathcal{S}, \mathcal{A}, \mathcal{P}, r, \gamma)$. At each time step $t$, the agent at state $S_t \in \mathcal{S}$ takes an action $A_t \in \mathcal{A}$ and the next state $S_{t+1} \in \mathcal{S}$ is determined by the transition probability kernel $\mathcal{P}(s'|s, a) = \Pr(S_{t+1} = s'|S_t = s, A_t = a)$. Then the environment generates an immediate reward $R_{t+1}$. The objective of the agent is to learn a policy $\pi : \mathcal{S} \times \mathcal{A} \rightarrow [0, 1]$ maximizing the expected discounted cumulative rewards $G_t = \mathbb{E}\left[\sum\limits_{k=0}^{\infty} \gamma^k R_{t+k+1}\right]$ where $\gamma \in [0, 1)$ is a discount factor.

### 2.1 BELLMAN EXPECTATION EQUATION.

Value function-based RL methods (Mnih et al., 2015; Hessel et al., 2018) aim to estimate the value function, which represents the expected return starting from a given state or state-action pair. The state-value function is defined as $V_\pi(s) = \mathbb{E}_\pi [G_t|S_t = s]$. For a stationary policy $\pi$, the state–value function satisfies

$$V_\pi(s) = \sum_{a \in \mathcal{A}} \pi(a \,|\, s) \sum_{s' \in \mathcal{S}} \mathcal{P}(s' \,|\, s, a) \big[ r(s, a, s') + \gamma \, V_\pi(s') \big], \tag{1}$$

which is also known as the Bellman expectation equation.

Consider a finite state set $\mathcal{S} = \{s_1, \ldots, s_{|\mathcal{S}|}\}$, so that the value function and transition operator admit a vector–matrix representation. Let $V_\pi = \left(V_\pi(s_1), \ldots, V_\pi(s_{|\mathcal{S}|})\right)^\top$, $r_\pi = \left(r_\pi(s_1), \ldots, r_\pi(s_{|\mathcal{S}|})\right)^\top$, and define the policy-dependent transition matrix $[P_\pi]_{ij} = \sum_{a \in \mathcal{A}} \pi(a \mid s_i)\, \mathcal{P}(s_j \mid s_i, a)$. Then Equation 1 can be formulated as the compact matrix form

$$\mathcal{T}V_\pi = r_\pi, \tag{2}$$

where $\mathcal{T} = I - \gamma P_\pi$ is the Bellman residual operator. For $\gamma < 1$, the closed-form solution is obtained as

$$V_\pi = (I - \gamma P_\pi)^{-1} r_\pi. \tag{3}$$

## 2.2 LAPLACIAN OF STATE TRANSITION GRAPHS.

The state transition graph of an MDP $(\mathcal{S}, \mathcal{A}, \mathcal{P}, r, \gamma)$ provides a structural representation of the environment by modeling the transitions between states as a graph (Şimşek & Barto, 2008). Formally, an state transition graph $G = (V, E, W)$ is a weighted directed graph where the vertices $V = \mathcal{S}$ correspond to the set of states and the directed edges $E \subseteq \mathcal{S} \times \mathcal{S}$ represent corresponding state transitions. An edge $(s, s') \in E$ exists if there exists an action $a \in \mathcal{A}$ such that the transition probability $\mathcal{P}(s'|s, a) > 0$. The weight function $W : E \rightarrow \mathbb{R}^+$ assigns each edge a positive value that captures the possibility of the corresponding transition.

Consider an MDP with a finite state set, we embed its transition dynamics in a weighted graph whose adjacency matrix is $W = [W_{ss'}]_{s,s' \in \mathcal{S}}$ and whose diagonal degree matrix is

$$D = \mathrm{diag}\big(d(s_1), \ldots, d(s_{|\mathcal{S}|})\big), \quad d(s) = \sum_{s'} W_{ss'}. \tag{4}$$

The topological structure of the graph is then characterized by the random-walk Laplacian

$$L = I - D^{-1}W, \tag{5}$$

whose eigenvectors associated with the smallest (non-zero) eigenvalues minimize the discrete Dirichlet energy and therefore vary smoothly within highly connected regions while changing sharply across narrow bridges (Von Luxburg, 2007; Belkin & Niyogi, 2003).

## 2.3 OPTION FRAMEWORK.

The option framework, introduced by (Sutton et al., 1999), extends RL by enabling agents to perform temporally extended actions, facilitating hierarchical decision-making. An option is defined as a tuple $\mathcal{O} = (\mathcal{I}_o, \pi_o, \beta_o)$, where $\mathcal{I}_o \subseteq \mathcal{S}$ is the initiation set specifying where the option can start, $\pi_o : \mathcal{S} \times \mathcal{A} \rightarrow [0, 1]$ is the intra-option policy governing behavior during the option's execution, and $\beta_o : \mathcal{S} \rightarrow [0, 1]$ is the termination condition defining the probability of ending the option at each state. Agents execute options by selecting an option, following its policy, and terminating with $\beta_o$.

# 3 OPTION DISCOVERY VIA VALUE POWER STRENGTH

Unlike traditional graph-based methods that compute bottleneck metrics with transition graphs, our approach leverages the structural analogy between the Bellman equation and Kirchhoff's current law to propose *Value Power Strength* (VPS)—a value function–based bottleneck metric. VPS enables the construction of intrinsic rewards that guide option learning toward or away from bottleneck regions.

## 3.1 VALUE POWER STRENGTH OF STATES

Consider the single-source single-sink resistance network illustrated in Figure 1. A reasonable bottleneck metric for node $s$ is the current flowing through it (Newman, 2005) as follows

$$I_i = \frac{1}{2} \sum_j W_{ij} |U_i - U_j|, \qquad for \quad i \neq s, g \tag{6}$$

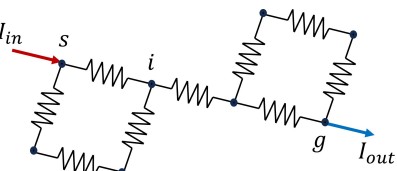

Figure 1: Single-source single-sink resistance network

where $W_{ij} = 1/R_{ij}$ denotes the conductance of edge $(i, j)$ (set to 0 if the edge is absent), $U_i$ is the electric potential at node $i$ with respect to the reference ground. The coefficient $1/2$ prevents each undirected edge from being counted twice. Specially, the current flow of source node and sink node is defined as $I_s = I_g = I_{in}$. The node current flow defined in Equation 6 essentially quantifies the share of the overall charge-transport task that is borne by a given vertex in the resistance network, and therefore serves as an effective indicator of bottlenecks with respect to current throughput.

The notion of a bottleneck, however, is not confined to flow; it also arises in energy conversion. If the network in Figure 1 is regarded not as a wire but as a heater that converts electrical energy into heat, then the node with the highest electrical power dissipation assumes the largest portion of the energy-conversion task and can likewise be viewed as an energy bottleneck. Following this idea, we propose to quantify the bottleneck property of a node by the electrical power it dissipates:

$$P_i = \frac{1}{2} \sum_j W_{ij}(U_i - U_j)^2. \tag{7}$$

Compared with the current flow metric Equation 6, the node power metric Equation 7 offers two main advantages: (i) it uniformly applies to all nodes, including the source and sink, as the external leads attached to $s$ and $g$ perform no energy conversion; (ii) its quadratic form $(U_i - U_j)^2$ is differentiable everywhere, facilitating smoother analysis and optimization.

More interestingly, the node power metric $P_i$ directly corresponds to a local decomposition of the graph Dirichlet energy. The (unnormalized) Dirichlet energy (Chung, 1997) of a potential field $U$ is

$$\mathcal{E}(U) = \frac{1}{2} \sum_{i,j} W_{ij}(U_i - U_j)^2 = \frac{1}{2} U^\top L U = \sum_i P_i, \tag{8}$$

where $L$ is the combinatorial Laplacian. Thus, nodes with large $P_i$ indicate regions where the global Dirichlet energy is concentrated.

To compute the node-power metric for every state directly within RL, we avoid explicit state-transition graph construction by leveraging the analogy between the Bellman equation and Kirchhoff's current law. Kirchhoff's law can be written as

$$LU = I, \tag{9}$$

where $U$ and $I$ are the node potential and external current vectors. $L$ is the graph Laplacian defined as $L = D - G$ where $G$ with elements $G_{ij} = W_{ij}$ and $D$ are the conductance matrix and the diagonal degree matrix, separately. Both the Bellman equation Equation 2 and Kirchhoff's law are discrete analogues of the Poisson equation,

$$\nabla^2 \phi = -\rho, \tag{10}$$

where $\phi$ is a potential and $\rho$ is a source term. In this analogy, the electric potential $U(s)$ is the energy released by moving a unit charge from node $s$ to ground, while the value function $V(s)$ is the expected discounted reward obtained from state $s$ to absorbing states. Thus, both $U(s)$ and $V(s)$ quantify the potential for future "energy" or "reward" gain, establishing a natural correspondence between the physical and RL domains. Consider Equation 2, Equation 9, Equation 10 and see Table 1 for a summary of these connections.

According to Table 1, the node power $P_i$ is mapped to a value function-based bottleneck metric defined as follows. The coefficient $1/2$ is omitted since it is shared by all states.

**Definition 1** (Value Power Strength). *Consider an MDP $(\mathcal{S}, \mathcal{A}, \mathcal{P}, r, \gamma)$ with a stationary policy $\pi$. The* Value Power Strength *(VPS) of state $s \in \mathcal{S}$ is defined as the expected squared difference in value*

| No. | Kirchhoff's Current Law | Bellman Equation | Poisson's Equation |
|---|---|---|---|
| 1 | External current input $I$ | External reward input $r_\pi$ | External source term $-\rho$ |
| 2 | Node voltage $U$ | State value $V_\pi$ | Scalar potential $\phi$ |
| 3 | Graph Laplacian operator $L$ | Bellman residual operator $\mathcal{T}$ | Laplacian operator $\nabla^2$ |
| 4 | Admittance matrix $G$ | State-transition matrix $P_\pi$ | — |

Table 1: Analogy between Kirchhoff's current law, Bellman equation, and Poisson's equation.

*between $s$ and its successor states:*

$$\varphi_\pi(s) = \mathbb{E}_{s' \sim P_\pi(\cdot | s)} \left[ (V_\pi(s) - V_\pi(s'))^2 \right]. \tag{11}$$

*When the state space $\mathcal{S}$ is finite, VPS can be formulated as*

$$\varphi_\pi(s) = \sum_{s' \in \mathcal{S}} P_\pi(s' \mid s) \left( V_\pi(s) - V_\pi(s') \right)^2 \tag{12}$$

*where $P_\pi(s' \mid s)$ is the policy-induced transition probability, given for discrete action spaces by $P_\pi(s' \mid s) = \sum_{a \in \mathcal{A}} \pi(a \mid s) \mathcal{P}(s' \mid s, a)$.*

According to Table 1, since the node power in Equation 7 measures how much responsibility node $i$ bears in the dissipation of power within a resistive network, VPS can be interpreted analogously as quantifying how much responsibility a state $s$ bears in the propagation of reward as the value function is formed through the Bellman equation. Thus, VPS essentially depicts the bottlenecks of reward diffusion in the state space.

### 3.2 ONLINE ESTIMATE OF VPS

Since the value of VPS depends on the value function $V_\pi$, we have proposed a simultaneous update law of $V_\pi$ and $\varphi_\pi$ as below.

$$\begin{cases} V_{t+1}(S_t) \leftarrow V_t(S_t) + \alpha_t \left[ R_t + \gamma V_t(S_{t+1}) - V_t(S_t) \right] \\ \varphi_{t+1}(S_t) \leftarrow \varphi_t(S_t) + \beta_t \left[ (V_t(S_t) - V_t(S_{t+1}))^2 - \varphi_t(S_t) \right] \end{cases} \tag{13}$$

The convergence of the update law is proved by Proposition 1 in Appendix A.1.

Although Equation 13 offers an online algorithm for VPS estimation, its definition still depends on a specific policy $\pi$ and reward $r$. To preliminarily validate the effectiveness of VPS for bottleneck identification, we consider a GridWorld environment "Rooms" which contains different types of bottlenecks (Figure 2). The agent starts from random positions, each episode has a fixed length, and follows a random walk policy with four primitive actions (up, down, left and right). For rewards, we examine: (i) assigning a reward of 1 to a few designated states (see "Value Power Strength" in Figure 2); and (ii) sequentially assigning a reward of 1 to each grid and aggregating the resulting VPS distributions (see "Cumulative Value Power Strength"). As shown in Figure 2, placing a reward near a bottleneck effectively highlights it via VPS, while the cumulative VPS closely matches the patterns of current flow and shortest path betweenness centrality (Brandes & Fleischer, 2005; Borgatti & Everett, 2006).

### 3.3 VPS-BASED OPTION DISCOVERY

Based on the above, VPS can reveal reward diffusion bottlenecks in MDPs under random walk policies with suitable reward design. However, two challenges remain for option discovery: (i) how to design rewards to estimate VPS; (ii) how to design options with VPS.

For the first challenge, we propose that the Gaussian noise reward is an efficient choice. For each random trial $m = 1, \ldots, M$, a Gaussian noise reward is assigned to each state $s \in \mathcal{S}$, independently sampled as

$$r^{(m)}(s) \sim \mathcal{N}(0, \sigma^2), \qquad \forall s \in \mathcal{S}, \tag{14}$$

where $\sigma^2$ is the variance. The rationality of this design will be revealed by the following theorem.

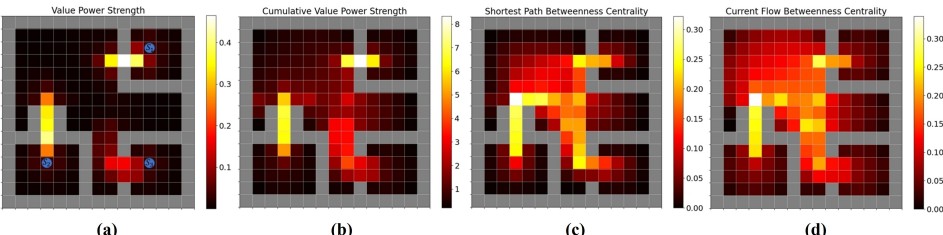

(a)          (b)          (c)          (d)

Figure 2: **VPS and betweenness centrality distribution in GridWorld-Rooms**: (a) VPS distribution when the agent performs a random walk and only three selected grids provide a visit reward of 1 (all other grids give reward 0). (b) Cumulative VPS map obtained by sequentially assigning a visit reward of 1 to each grid (one target grid per experiment, all others 0), computing VPS for each case, and aggregating the resulting VPS fields. (c) Shortest-path betweenness centrality. (d) Current-flow betweenness centrality.

**Lemma 1.** *Let $(\mathcal{S}, \mathcal{A}, \mathcal{P}, r, \gamma)$ be a finite MDP with stationary policy $\pi$ such that the induced Markov chain is reversible. Let $P_\pi \in \mathbb{R}^{|\mathcal{S}| \times |\mathcal{S}|}$ denote the symmetric policy-induced transition matrix, and define the random walk Laplacian as $L = I - P_\pi$. The Bellman operator is denoted by $\mathcal{T} = I - \gamma P_\pi$ with $0 < \gamma < 1$. Then, for every eigenpair $(\mu_k, v_k)$ of $L$, s.t., $Lv_k = \mu_k v_k$, the vector $v_k$ is also an eigenvector of $\mathcal{T}$, with corresponding eigenvalue $\lambda_k = 1 - \gamma(1 - \mu_k)$.*

*Proof.* See Appendix A.2. □

**Theorem 1** (Spectral Solution of the State-Value Function)**.** *Consider a finite, reversible MDP as in Lemma 1, and let $P_\pi$ be the policy-induced transition matrix with unique stationary distribution $\mathbf{d} \in \mathbb{R}^{|\mathcal{S}|}$ (i.e., $P_\pi^\top \mathbf{d} = \mathbf{d}$, $\sum_i d_i = 1$, $d_i > 0$). Define the random walk Laplacian as $L = I - D^{-1}W = I - P_\pi$, where $D = \mathrm{diag}(\mathbf{d})$. Let $\{(\mu_k, v_k)\}_{k=1}^{|\mathcal{S}|}$ be the eigenpairs of $L$ with $\{v_k\}$ forming an orthonormal basis under the $\mathbf{d}$-weighted inner product:*

$$\langle f, g \rangle_{\mathbf{d}} := \sum_{i=1}^{|\mathcal{S}|} d_i \, f(s_i) \, g(s_i).$$

*Let $\mathbf{r} \in \mathbb{R}^{|\mathcal{S}|}$ denote the reward vector whose $i$-th entry $\mathbf{r}_i$ is the expected immediate reward at $s_i$. Then, the solution $V$ to the Bellman equation $\mathcal{T}V = \mathbf{r}$ admits the following spectral decomposition:*

$$V = \sum_{k=1}^{|\mathcal{S}|} \frac{1}{1 - \gamma(1 - \mu_k)} \, \langle \mathbf{r}, v_k \rangle_{\mathbf{d}} \, v_k. \tag{15}$$

*Proof.* See Appendix A.3. □

The use of Gaussian noise rewards for VPS is motivated by the spectral result of Theorem 1. Since the Gaussian reward vector $\mathbf{r}$ in Equation 14 is isotropic, it does not bias any eigenvector $v_k$ of the random walk Laplacian $L$ in the $\mathbf{d}$-weighted inner product space, and the inner product $\langle \mathbf{r}, v_k \rangle_{\mathbf{d}}$ is uniformly distributed in expectation. Consequently, the state-value function $V$ is dominated by low-frequency components—those with small $\mu_k$—due to the spectral scaling $[1 - \gamma(1 - \mu_k)]^{-1}$, while high-frequency eigenvectors are suppressed. This ensures $V$ reflects the large-scale structure of the state space, resulting in a smooth partitioning (Chung, 1997). Note that the analysis of Theorem 1 strictly holds only for reversible MDPs; it still provides useful guidance for reward design in VPS.

For the second challenge, inspired by the Eigenoption (Machado et al., 2017b), the VPS-based dual potential-difference intrinsic rewards $r_{\text{int}}^{(m)}(s', s) = \pm \big( \varphi^{(m)}(s') - \varphi^{(m)}(s) \big)$ are considered to obtain options towards or away from bottlenecks, where the superscript $(m)$ denotes the $m$-th pair of options corresponding to the $m$-th random reward sampling.

The $m$-th pair of options is constructed by maximizing the expected discounted cumulative intrinsic reward $r_{\text{int}}^{(m)}(s', s)$, with the intra-option policy $\pi_o$ and termination rule $\beta_o$ parameterized for each

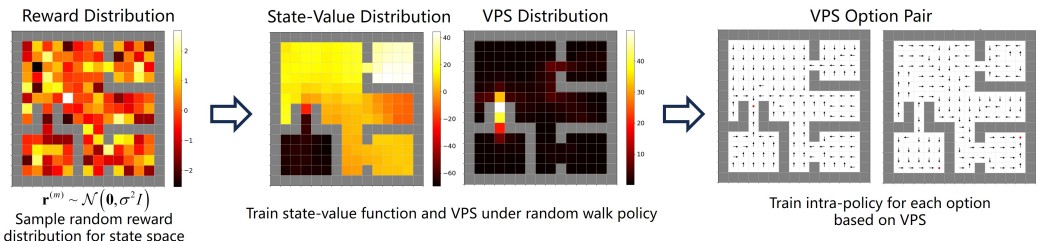

Figure 3: Pipeline for VPS-based option discovery

option $o$. The optimal option-value function is defined as

$$Q_o^*(s,a) = \max_{\pi_o} \mathbb{E}\left[\sum_{t=0}^{\infty} \gamma^t r_{\text{int}}^{(m)}(S_{t+1}, S_t) \,\Big|\, S_0 = s, A_0 = a, o\right]. \tag{16}$$

The corresponding greedy intra-option policy is given by $a_o^*(s) = \arg\max_{a\in\mathcal{A}} Q_o^*(s,a)$.

The initiation set of each option is set as the entire state space $\mathcal{S}$. The termination probability $\beta_o(s)$ is defined as

$$\beta_o(s) = \begin{cases} 1, & \text{if } Q_o^*(s,a) < 0, \quad for \quad \forall a \in \mathcal{A} \\ 1/N, & \text{otherwise} \end{cases} \tag{17}$$

where $N$ is a manually specified positive constant. The options designed in this manner enable the agent to reach states with the highest or lowest VPS values (or local extrema, depending on the discount factor).

Figure 3 illustrates the overall procedure of VPS-based option discovery, while the algorithmic procedure is provided in Appendix B.

### 3.4 FUNCTION APPROXIMATION CASES

To generalize VPS to high-dimensional or continuous state spaces, we learn a shared encoder $z_\phi^s \in \mathbb{R}^d$ with a value head $V_\theta$ and a VPS head $\varphi_\psi$. We train $V_\theta$ with a target network $V_{\bar\theta}$ and estimate VPS as a conditional regression target built from $V_{\bar\theta}$:

$$\mathcal{L}_V(\theta,\phi) = \mathbb{E}_{(s,r,s')\sim\mathcal{D}}\big[\,r + \gamma\,\bar{V}_{\bar\theta}(z_\phi^{s'}) - V_\theta(z_\phi^s)\,\big]^2, \tag{18}$$

$$\mathcal{L}_\varphi(\psi,\phi) = \mathbb{E}_{(s,s')\sim\mathcal{D}}\big[\,\big(V_{\bar\theta}(z_\phi^s) - V_{\bar\theta}(z_\phi^{s'})\big)^2 - \varphi_\psi(z_\phi^s)\,\big]^2. \tag{19}$$

In practice, we optionally stop the gradient from $\mathcal{L}_\varphi$ to $z_\phi$ if training is unstable.

In high-dimensional or continuous spaces, we construct VPS-driving random rewards using Random Fourier Features (RFF) (Rahimi & Recht, 2007) on the learned representation $z_\phi^s$. Let $k(z,z') = \exp\big(-\|z-z'\|^2/(2\ell^2)\big)$ be the RBF kernel with length-scale $\ell > 0$. We sample

$$\mathbf{w}_j \sim \mathcal{N}(0, \ell^{-2}I_d), \qquad b_j \sim \text{Uniform}(0, 2\pi),$$

and define the $K$-dimensional randomized feature map

$$\zeta_j(s) \;=\; \sqrt{\tfrac{2}{K}}\;\cos\big(\mathbf{w}_j^\top z(s) + b_j\big), \quad j = 1,\ldots,K.$$

The $m$-th random reward is a random linear combination of these features,

$$r^{(m)}(s) \;=\; \sum_{j=1}^{K} a_j^{(m)}\,\zeta_j(s), \qquad a_j^{(m)} \sim \mathcal{N}(0,1)\ \text{ i.i.d.} \tag{20}$$

As $K \to \infty$, $r^{(m)}$ converges (in mean-square) to a sample from a zero-mean Gaussian process $\mathcal{GP}(0,k)$ with the kernel $k$ specified above.

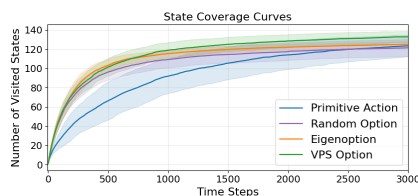

Figure 4: State coverage curve for Gridworld-Rooms

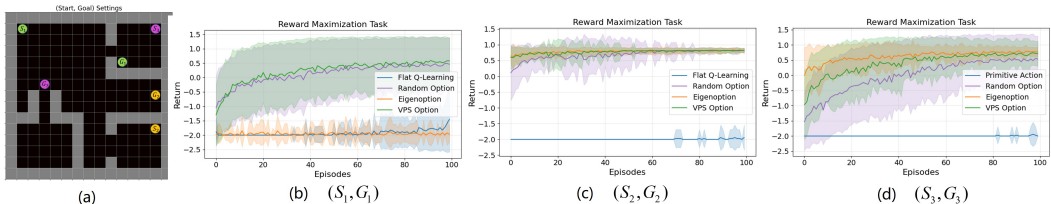

Figure 5: Return curves for different (start, goal) pairs in GridWorld-Rooms

We explicitly build the RFF rewards on the representation $z(s)$ so that the induced random fields are continuous in $z$. In many control domains, representation learning (e.g., predictive or contrastive training) aligns $z$ with the underlying transition dynamics, making states that are close in the transition graph tend also to be close in the Euclidean geometry of $z$. Consequently, continuity in $z$ serves as a practical proxy for smoothness on the state-transition graph, which is desirable for estimating VPS. Designing random rewards that match the transition geometry more faithfully in continuous, high-dimensional tasks (e.g., using successor-feature or diffusion kernels on $z$) is an important direction for future work.

## 4 EXPERIMENTAL RESULTS

We conduct experiments on classic tabular domains, Visual GridWorld, and Atari 2600 environments. In all experiments, agents collect transitions under a random walk policy, and VPS is trained with random rewards Equation 14 and Equation 20.

### 4.1 TABULAR CASES

For the tabular cases, we evaluate on GridWorld-Rooms (Figure 2) and Gym Taxi-v3 (Brockman et al., 2016). Each VPS option set comprises 20 options from 10 random rewards, and we compare against 20 eigenoptions (from the 10 smallest Laplacian eigenvalues) and 20 Random Options (policies trained with random potential intrinsic rewards). All option discovery methods are repeated with 10 different random seeds, resulting in 10 independent sets of options. Each set of options is evaluated over 10 random seeds for both exploration and reward collection tasks. Additional experimental details are provided in Appendix C.1.

For GridWorld-Rooms, as shown in Figure 4, VPS options yield higher exploration coverage than primitive actions, random options, and even eigenoptions. Further, Q-learning with VPS options on various start–goal pairs in Figure 5 confirms that VPS options help agents traverse bottlenecks and acquire rewards more effectively.

In the Taxi-v3 environment, we employ the same experimental protocol as in GridWorld-Rooms (10 option sets × 10 training runs). Interestingly, a considerable proportion of VPS options are able to complete the pick-up and drop-off task without ever receiving external rewards during training, as shown in Figure 6(a). This phenomenon arises because bottleneck states—where the taxi, passenger, and destination coincide—are automatically identified. Figures 6(b) and 6(c) report the frequency of automatic task completion and reward acquisition via Q-learning. Overall, VPS options effectively capture meaningful behaviors and facilitate reward acquisition, although adding options accelerates learning but does not always improve final performance.

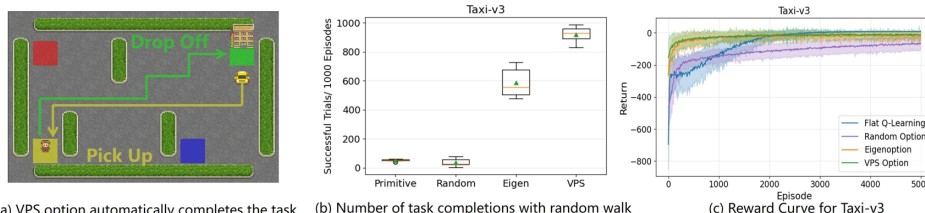

(a) VPS option automatically completes the task    (b) Number of task completions with random walk    (c) Reward Curve for Taxi-v3

Figure 6: **Performance of different options in Taxi-v3**: A "successful trial" refers to an episode in which the taxi successfully transports a passenger from the starting point to the destination once.

## 4.2 VISUAL GRIDWORLD AND ATARI 2600

To verify the applicability of VPS options in high-dimensional state spaces, we consider a visual GridWorld-Corridor environment shown in Figure 7 based on Minigrid (Chevalier-Boisvert et al., 2023), where the agent observes 84×84 grayscale images. Using random walks with four actions (up, down, left and right) and RFF-based rewards, we train neural networks to approximate the value function, VPS, and DQN-based options. Figure 7 shows that the VPS option framework effectively generalizes to this high-dimensional setting via function approximation.

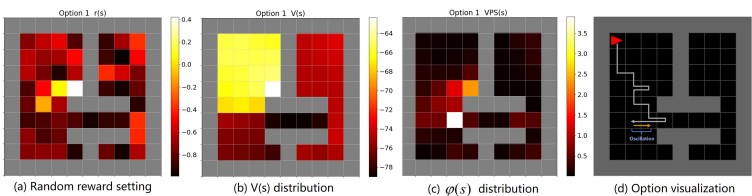

(a) Random reward setting    (b) V(s) distribution    (c) $\varphi(s)$ distribution    (d) Option visualization

Figure 7: VPS option training process for visual GridWorld-Rooms

Furthermore, in Atari 2600 games, we let the agent perform random walks to collect transition data, and train 8 different VPS with RFF rewards using single-frame 84×84 grayscale images as input. We then evaluate human gameplay videos by computing the $\varphi$ value of each frame using the trained VPS networks. Figure 8 illustrates that in the ALE/Venture-v5 environment, VPS peaks reliably correspond to interpretable bottleneck frames—specifically, doorways between different rooms (the positions of the agent are marked by white cross stars). Additional results for more Atari games are provided in Appendix C.2.

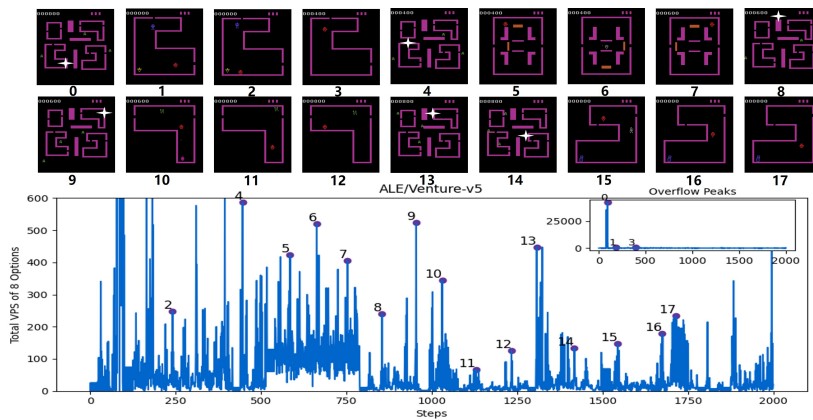

Figure 8: Bottlenecks identified through VPS in ALE/Venture-v5

## 5 RELATED WORK

A central observation behind VPS is that it measures the local Dirichlet energy of value functions on the state-transition graph, which naturally connects our method to Laplacian representation learning that recovers MDP geometry from the Laplacian spectrum. Early work on proto-value functions (PVFs) (Mahadevan, 2005; Mahadevan & Maggioni, 2007) used Laplacian eigenvectors as basis functions, showing that the spectrum compactly encodes global structure useful for control.

Building on this spectral view, eigenoptions (Machado et al., 2017a) define options from Laplacian eigenvectors—directions of minimal Dirichlet energy—to induce smooth large-scale transitions and improve exploration. Subsequent work clarified the link to successor representations (Machado et al., 2017b; 2023), enabling eigenoption-style behaviors without explicit eigendecomposition.

Recent advances make Laplacian methods scalable in high-dimensional and continuous domains via improved Laplacian learning objectives (Wu et al., 2019; Gomez et al., 2024) and more general spectral operator learning (Touati et al., 2023; Ryu et al., 2025). These representations support online spectral option discovery such as Deep Covering Eigenoptions (Klissarov & Machado, 2023) and have also been shown to aid credit assignment (Kotamreddy & Machado, 2025).

However, Laplacian eigenvectors encode global smoothness rather than bottleneck structure specifically. While coarse bottlenecks may appear in components such as the Fiedler eigenvector, others emphasize room centers, edges, or other geometry, and it is unclear a priori which eigenvectors correspond to bottlenecks at which scales. Thus eigenoption-based methods can mix heterogeneous structures. In contrast, VPS directly measures a state's reward-diffusion "responsibility," producing explicitly bottleneck-oriented options with stronger interpretability when narrow passages dominate task difficulty.

VPS is also related to methods that learn many auxiliary value functions from random cumulants. Random-cumulant General Value Functions and auxiliary tasks use multiple cumulants as self-supervised signals for representation learning and exploration, where each cumulant defines a separate value head (Lyle et al., 2021; Zheng et al., 2021; Ramesh et al., 2022). Proto-Value Networks extend this idea to hundreds of procedurally defined auxiliary value functions; in the limit of infinitely many Gaussian random cumulants, the subspace spanned by these value predictions converges to the leading eigenspaces of the successor/Laplacian operator (Jesse Farebrother, 2023). In these works, random cumulants primarily enrich shared representations for downstream control. By contrast, our use of multiple random rewards is diagnostic: we aggregate the squared value differences induced by each reward into a single VPS scalar, interpret it as node power in an equivalent electrical network, and use it as a targeted bottleneck indicator to construct options that explicitly steer the agent toward or away from high-VPS states, rather than treating the auxiliary values themselves as generic features.

## 6 CONCLUSION

This paper introduces Value Power Strength (VPS), a value function–based metric for identifying reward diffusion bottlenecks and discovering interpretable options in RL. By formalizing the analogy between the Bellman equation and Kirchhoff's current law, VPS enables principled bottleneck identification directly from value estimates, without requiring explicit state-transition graph construction. Comprehensive experiments across different environments demonstrate that VPS options consistently identify semantically meaningful subgoals and improve the exploration efficiency. For future work, we aim to design suitable high-level policies to further evaluate VPS options on more complex tasks, and explore the use of VPS as an intrinsic reward to enhance exploration in challenging RL domains.

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

# Appendix

## A. Proofs of Main Results

### A.1. Proposition 1 and its proof

$$
\begin{cases}
V_{t+1}(S_t) \leftarrow V_t(S_t) + \alpha_t \left[ R_t + \gamma V_t(S_{t+1}) - V_t(S_t) \right] \\
\varphi_{t+1}(S_t) \leftarrow \varphi_t(S_t) + \beta_t \left[ (V_t(S_t) - V_t(S_{t+1}))^2 - \varphi_t(S_t) \right]
\end{cases}
\tag{21}
$$

**Proposition 1.** *Consider a finite MDP defined by the tuple $(\mathcal{S}, \mathcal{A}, \mathcal{P}, r, \gamma)$ with a stationary policy $\pi$. Let $\{V_t\}_{t \geq 0}$ and $\{\varphi_t\}_{t \geq 0}$ be the sequences of value estimates and VPS estimates updated according to Equation 21. Suppose the following conditions hold:*

1. *The Markov chain induced by $\pi$ is ergodic (irreducible and aperiodic);*

2. *The learning rates $\{\alpha_t\}$ and $\{\beta_t\}$ satisfy the Robbins-Monro conditions:*

$$
\sum_t \alpha_t = \sum_t \beta_t = \infty, \quad \sum_t \alpha_t^2 < \infty, \quad \sum_t \beta_t^2 < \infty;
$$

3. *The second moment of the reward is uniformly bounded:*

$$
\mathbb{E}_\pi \left[ R_t^2 \mid S_t = s \right] \leq R_{\max}^2 < \infty, \quad \forall s \in \mathcal{S}.
$$

*Then, for $\forall s \in \mathcal{S}$, the sequences $\{V_t\}$ and $\{\varphi_t\}$ converge almost surely to the true state-value function $V_\pi$ and the true VPS function $\varphi_\pi$, respectively.*

*Proof.* The convergence of the value function sequence $\{V_t\}$ to $V_\pi$ under the given conditions is a well-established result (Tsitsiklis & Van Roy, 1996). We thus focus on proving the almost sure convergence of the VPS sequence $\{\varphi_t\}$ to $\varphi_\pi$.

**Step 1: Preliminaries and Notation**

Since the Markov chain induced by $\pi$ is ergodic, every state $s \in \mathcal{S}$ is visited infinitely often almost surely. Let $\{\tau_k(s)\}_{k=1}^\infty$ denote the increasing sequence of time steps at which state $s$ is visited. Consider the subsequence $\{\varphi_{\tau_k}(s)\}_{k \geq 1}$.

It is standard that, under the Robbins-Monro conditions and the assumptions on rewards, $V_t(s) \rightarrow V_\pi(s)$ almost surely for all $s \in \mathcal{S}$. Since $\mathcal{S}$ is finite, this convergence is uniform:

$$
\forall \varepsilon > 0, \ \exists K(\varepsilon) \text{ such that } \forall k \geq K(\varepsilon), \ \max_{s \in \mathcal{S}} |V_{\tau_k}(s) - V_\pi(s)| < \varepsilon \quad \text{a.s.}
\tag{22}
$$

**Step 2: Stochastic Update for VPS**

By definition, when $s$ is visited at time $\tau_k$, the update is:

$$
\varphi_{\tau_k + 1}(s) = \varphi_{\tau_k}(s) + \beta_{\tau_k} \left[ (V_{\tau_k}(s) - V_{\tau_k}(S_{\tau_k + 1}))^2 - \varphi_{\tau_k}(s) \right]
$$
$$
= (1 - \beta_{\tau_k}) \varphi_{\tau_k}(s) + \beta_{\tau_k} (V_{\tau_k}(s) - V_{\tau_k}(S_{\tau_k + 1}))^2.
$$

Let $e_k(s) := \varphi_{\tau_k}(s) - \varphi_\pi(s)$ denote the error at the $k$-th visit to $s$. We have

$$
e_{k+1}(s) = (1 - \beta_{\tau_k}) e_k(s) + \beta_{\tau_k} \left[ (V_{\tau_k}(s) - V_{\tau_k}(S_{\tau_k + 1}))^2 - \varphi_\pi(s) \right]
$$
$$
= (1 - \beta_{\tau_k}) e_k(s) + \beta_{\tau_k} \left( A_k + B_k \right)
\tag{23}
$$

where we decompose

$$
A_k := (V_{\tau_k}(s) - V_{\tau_k}(S_{\tau_k + 1}))^2 - (V_\pi(s) - V_\pi(S_{\tau_k + 1}))^2,
$$
$$
B_k := (V_\pi(s) - V_\pi(S_{\tau_k + 1}))^2 - \mathbb{E}_{s' \sim P_\pi(\cdot | s)} \left[ (V_\pi(s) - V_\pi(s'))^2 \right].
$$

**Step 3: Uniform Control of the Bias Term $A_k$**

Let $\delta_1 = V_{\tau_k}(s) - V_\pi(s)$, $\delta_2 = V_{\tau_k}(S_{\tau_k+1}) - V_\pi(S_{\tau_k+1})$. Then,

$$
\begin{aligned}
A_k &= (V_{\tau_k}(s) - V_{\tau_k}(S_{\tau_k+1}))^2 - (V_\pi(s) - V_\pi(S_{\tau_k+1}))^2 \\
&= \left((V_\pi(s) + \delta_1) - (V_\pi(S_{\tau_k+1}) + \delta_2)\right)^2 - (V_\pi(s) - V_\pi(S_{\tau_k+1}))^2 \\
&= \left((V_\pi(s) - V_\pi(S_{\tau_k+1})) + (\delta_1 - \delta_2)\right)^2 - (V_\pi(s) - V_\pi(S_{\tau_k+1}))^2 \\
&= 2(V_\pi(s) - V_\pi(S_{\tau_k+1}))(\delta_1 - \delta_2) + (\delta_1 - \delta_2)^2.
\end{aligned}
$$

Applying triangle inequality and uniform convergence Equation 22, when $k$ is large enough,

$$
|\delta_1|, |\delta_2| < \varepsilon, \quad |V_\pi(s) - V_\pi(S_{\tau_k+1})| \le 2M,
$$

where $M = \max_{s \in \mathcal{S}} |V_\pi(s)|$. Thus,

$$
|A_k| \le 2 \cdot 2M \cdot 2\varepsilon + (2\varepsilon)^2 = 8M\varepsilon + 4\varepsilon^2.
$$

Therefore, $A_k \to 0$ almost surely as $k \to \infty$.

**Step 4: Martingale Difference Property of $B_k$**

For each $k$, $B_k$ is a martingale difference: conditioned on $\mathcal{F}_{\tau_k}$ (the $\sigma$-algebra up to time $\tau_k$),

$$
\mathbb{E}[B_k \mid \mathcal{F}_{\tau_k}] = 0.
$$

Moreover, since $|V_\pi(s)| \le M$, it follows $|B_k| \le (2M)^2 = 4M^2$ almost surely, so the sequence has uniformly bounded second moment.

**Step 5: Convergence via Auxiliary Sequence Construction**

The error dynamics in Equation 23 can be expressed as:

$$
e_{k+1}(s) = (1 - \beta_{\tau_k})e_k(s) + \beta_{\tau_k}w_k + \beta_{\tau_k}\delta_k \tag{24}
$$

where:

$$
\begin{aligned}
w_k &:= B_k + (A_k - \mathbb{E}[A_k \mid \mathcal{F}_{\tau_k}]) \\
\delta_k &:= \mathbb{E}[A_k \mid \mathcal{F}_{\tau_k}]
\end{aligned}
$$

By construction, $w_k$ satisfies:

(a) $\mathbb{E}[w_k \mid \mathcal{F}_{\tau_k}] = 0$        (martingale difference)

(b) $|w_k| \le |B_k| + |A_k| + \mathbb{E}[|A_k| \mid \mathcal{F}_{\tau_k}] \le 4M^2 + 16M\varepsilon + 8\varepsilon^2 < \infty$ a.s.

Thus $\mathbb{E}[w_k^2 \mid \mathcal{F}_{\tau_k}] \le C$ for some $C < \infty$ almost surely.

Consider the auxiliary sequence defined by:

$$
\widetilde{e}_{k+1}(s) = (1 - \beta_{\tau_k})\widetilde{e}_k(s) + \beta_{\tau_k}w_k, \quad \widetilde{e}_0(s) = e_0(s) \tag{25}
$$

To apply Lemma 1 in (Tsitsiklis & Van Roy, 1996), the following conditions should be satisfied:

(a) $\mathbb{E}[w_k \mid \mathcal{F}_{\tau_k}] = 0$

(b) $\mathbb{E}[w_k^2 \mid \mathcal{F}_{\tau_k}] \le C < \infty$

(c) $\beta_{\tau_k} \in [0, 1]$ (by assumption)

(d) $\sum_{k=0}^{\infty} \beta_{\tau_k} = \infty$ (Robbins-Monro)

(e) $\sum_{k=0}^{\infty} \beta_{\tau_k}^2 < \infty$ (Robbins-Monro)

While Lemma 1 requires $\beta_{\tau_k} \in [0, 1]$, our sequence may have $\beta_{\tau_k} > 1$ for some $k$. However, since $\sum \beta_{\tau_k}^2 < \infty$, we have $\beta_{\tau_k} \to 0$ a.s. Thus for any $\epsilon > 0$, there exists $K_\epsilon$ such that for $k \geq K_\epsilon$, $\beta_{\tau_k} < \epsilon$. We can choose $\epsilon < 1$ so that for $k \geq K_\epsilon$, $\beta_{\tau_k} \in [0, 1]$.

For $k < K_\epsilon$, the finite number of updates where $\beta_{\tau_k} \geq 1$ do not affect almost sure convergence. Specifically, we can restart the sequence at $k = K_\epsilon$ with initial condition $e_{K_\epsilon}(s)$, which is bounded a.s. by the finite state space assumption. Therefore, without loss of generality, we assume $\beta_{\tau_k} \in [0, 1]$ for all $k$.

Under these conditions, Lemma 1 implies $\widetilde{e}_k(s) \to 0$ almost surely as $k \to \infty$.

The difference between the sequences satisfies:

$$e_k(s) - \widetilde{e}_k(s) = \sum_{m=0}^{k-1} \beta_{\tau_m} \delta_m \prod_{j=m+1}^{k-1} (1 - \beta_{\tau_j})$$

$$\leq \sup_{0 \leq m \leq k-1} |\delta_m| \cdot \underbrace{\sum_{m=0}^{k-1} \beta_{\tau_m} \prod_{j=m+1}^{k-1} (1 - \beta_{\tau_j})}_{H_k}$$

The sum $H_k$ satisfies:

$$H_k = \sum_{m=0}^{k-1} \beta_{\tau_m} \prod_{j=m+1}^{k-1} (1 - \beta_{\tau_j})$$

$$= 1 - \prod_{j=0}^{k-1} (1 - \beta_{\tau_j}) \quad \text{(telescoping sum identity)}$$

$$\leq 1$$

Therefore:

$$|e_k(s) - \widetilde{e}_k(s)| \leq \sup_{0 \leq m \leq k-1} |\delta_m|$$

Since $\delta_m \to 0$ almost surely (from Step 3), we have:

$$\lim_{k \to \infty} \sup_{0 \leq m \leq k-1} |\delta_m| = 0 \quad \text{a.s.}$$

Thus $|e_k(s) - \widetilde{e}_k(s)| \to 0$ almost surely.

Combining with $\widetilde{e}_k(s) \to 0$ a.s., we conclude $e_k(s) \to 0$ almost surely.

**Step 6: Extension to the Full Sequence**

The above argument holds for each $s \in \mathcal{S}$. Since the state space is finite and $\varphi_t(s)$ is only updated at visits to $s$, the entire sequence $\{\varphi_t(s)\}_{t \geq 0}$ converges almost surely to $\varphi_\pi(s)$. This completes the proof. $\square$

## A.2. Proof of Lemma 1

**Lemma 1.** *Let* $(\mathcal{S}, \mathcal{A}, \mathcal{P}, r, \gamma)$ *be a finite MDP with stationary policy* $\pi$ *such that the induced Markov chain is reversible. Let* $P_\pi \in \mathbb{R}^{|\mathcal{S}| \times |\mathcal{S}|}$ *denote the symmetric policy-induced transition matrix, and define the random walk Laplacian as* $L = I - P_\pi$. *The Bellman operator is denoted by* $\mathcal{T} = I - \gamma P_\pi$ *with* $0 < \gamma < 1$. *Then, for every eigenpair* $(\mu_k, v_k)$ *of* $L$, *s.t.,* $Lv_k = \mu_k v_k$, *the vector* $v_k$ *is also an eigenvector of* $\mathcal{T}$, *with corresponding eigenvalue* $\lambda_k = 1 - \gamma(1 - \mu_k)$.

*Proof.* Let $(\mu_k, v_k)$ be an eigenpair of the random walk Laplacian $L$, satisfying:

$$Lv_k = \mu_k v_k$$

By definition $L = I - P_\pi$, we have:

$$(I - P_\pi)v_k = \mu_k v_k \tag{26}$$

Rearranging terms yields:

$$P_\pi v_k = (1 - \mu_k)v_k \tag{27}$$

Now consider the Bellman operator $\mathcal{T} = I - \gamma P_\pi$:

$$\mathcal{T}v_k = (I - \gamma P_\pi)v_k = v_k - \gamma P_\pi v_k$$

Substituting Equation 27:

$$\mathcal{T}v_k = v_k - \gamma(1 - \mu_k)v_k = [1 - \gamma(1 - \mu_k)]\, v_k$$

Thus, $v_k$ is an eigenvector of $\mathcal{T}$ with eigenvalue $\lambda_k = 1 - \gamma(1 - \mu_k)$. $\qquad\square$

## A.3. Proof of Theorem 1

**Theorem 1** (Spectral Solution of the State-Value Function). *Consider a finite, reversible MDP as in Lemma 1, and let $P_\pi$ be the policy-induced transition matrix with unique stationary distribution $\mathbf{d} \in \mathbb{R}^{|\mathcal{S}|}$ (i.e., $P_\pi^\top \mathbf{d} = \mathbf{d}$, $\sum_i d_i = 1$, $d_i > 0$). Define the random walk Laplacian as $L = I - D^{-1}W = I - P_\pi$, where $D = \mathrm{diag}(\mathbf{d})$. Let $\{(\mu_k, v_k)\}_{k=1}^{|\mathcal{S}|}$ be the eigenpairs of $L$ with $\{v_k\}$ forming an orthonormal basis under the $\mathbf{d}$-weighted inner product:*

$$\langle f, g \rangle_{\mathbf{d}} := \sum_{i=1}^{|\mathcal{S}|} d_i \, f(s_i) \, g(s_i).$$

*Let $\mathbf{r} \in \mathbb{R}^{|\mathcal{S}|}$ denote the reward vector whose $i$-th entry $\mathbf{r}_i$ is the expected immediate reward at $s_i$. Then, the solution $V$ to the Bellman equation $\mathcal{T}V = \mathbf{r}$ admits the following spectral decomposition:*

$$V = \sum_{k=1}^{|\mathcal{S}|} \frac{1}{1 - \gamma(1 - \mu_k)} \, \langle \mathbf{r}, v_k \rangle_{\mathbf{d}} \, v_k. \tag{28}$$

*Proof.* The Bellman equation for the state-value function is given by:

$$\mathcal{T}V = \mathbf{r}$$

where $\mathcal{T} = I - \gamma P_\pi$ is the Bellman operator. This can be rewritten as:

$$(I - \gamma P_\pi)V = \mathbf{r} \tag{29}$$

By Lemma 1, for each eigenpair $(\mu_k, v_k)$ of $L = I - P_\pi$, $v_k$ is also an eigenvector of $\mathcal{T}$ with eigenvalue:

$$\lambda_k = 1 - \gamma(1 - \mu_k)$$

Since $P_\pi$ is symmetric and reversible, $L$ is symmetric under the $\mathbf{d}$-weighted inner product:

$$\langle Lu, v \rangle_{\mathbf{d}} = \langle u, Lv \rangle_{\mathbf{d}}$$

Thus $\{v_k\}$ forms an orthonormal basis for $\mathbb{R}^{|\mathcal{S}|}$ under $\langle \cdot, \cdot \rangle_{\mathbf{d}}$:

$$\langle v_i, v_j \rangle_{\mathbf{d}} = \delta_{ij}$$

Expand $V$ and $\mathbf{r}$ in this basis:

$$V = \sum_{k=1}^{|\mathcal{S}|} a_k v_k \tag{30}$$

$$\mathbf{r} = \sum_{k=1}^{|\mathcal{S}|} b_k v_k \tag{31}$$

where coefficients are given by:

$$a_k = \langle V, v_k \rangle_{\mathbf{d}}, \quad b_k = \langle \mathbf{r}, v_k \rangle_{\mathbf{d}}$$

Substitute into Equation 29:

$$(I - \gamma P_\pi) \left( \sum_{k=1}^{|\mathcal{S}|} a_k v_k \right) = \sum_{k=1}^{|\mathcal{S}|} b_k v_k$$

Using the eigenvalue relation from Lemma 1:

$$\sum_{k=1}^{|\mathcal{S}|} a_k \lambda_k v_k = \sum_{k=1}^{|\mathcal{S}|} b_k v_k$$

By orthogonality of $\{v_k\}$, we equate coefficients:

$$a_k \lambda_k = b_k \quad \text{for each } k$$

Thus:

$$a_k = \frac{b_k}{\lambda_k} = \frac{\langle \mathbf{r}, v_k \rangle_{\mathbf{d}}}{1 - \gamma(1 - \mu_k)}$$

Reconstructing $V$:

$$V = \sum_{k=1}^{|\mathcal{S}|} \frac{\langle \mathbf{r}, v_k \rangle_{\mathbf{d}}}{1 - \gamma(1 - \mu_k)} v_k$$

$\square$

## B. Implementation of VPS-Based Option Discovery

---

**Algorithm 1** Online VPS Option Discovery

---

**Require:** Number of random reward groups $k$, number of options $2k$, buffer size $B$, Gaussian reward variance $\sigma^2$, learning rates $\alpha$, $\beta$, Q-learning parameters, discount $\gamma$
 1: **Initialize** replay buffer $\mathcal{D} \leftarrow \emptyset$
 2: **for** each episode **do**
 3:    Collect transitions $(s, a, s')$ under random policy; store in $\mathcal{D}$ until $|\mathcal{D}| \geq B$
 4: **end for**
 5: **// Stage 1: VPS Estimation**
 6: **for** each $i = 1$ to $k$ **do**
 7:    Assign random Gaussian reward $r^{(i)}(s) \sim \mathcal{N}(0, \sigma^2)$ independently for all $s$
 8:    Initialize $V^{(i)}(s)$, $\varphi^{(i)}(s)$ arbitrarily for all $s$
 9:    **repeat**
10:       **for** each sampled transition $(s, a, s') \sim \mathcal{D}$ **do**
11:          $V^{(i)}(s) \leftarrow V^{(i)}(s) + \alpha\big[r^{(i)}(s) + \gamma V^{(i)}(s') - V^{(i)}(s)\big]$
12:          $\varphi^{(i)}(s) \leftarrow \varphi^{(i)}(s) + \beta\big[(V^{(i)}(s) - V^{(i)}(s'))^2 - \varphi^{(i)}(s)\big]$
13:       **end for**
14:    **until** $V^{(i)}$ and $\varphi^{(i)}$ converge for all $s$
15: **end for**
16: **// Stage 2: Option Q-Learning (dual directions)**
17: **for** each $i = 1$ to $k$ **do**
18:    **for** each sign $\xi \in \{+1, -1\}$ **do**
19:       Define intrinsic reward: $r_{\text{int}}^{(i,\xi)}(s, s') = \xi\left(\varphi^{(i)}(s') - \varphi^{(i)}(s)\right)$
20:       Initialize $Q_{i,\xi}(s, a)$ arbitrarily for all $s, a$
21:       **repeat**
22:          **for** each sampled transition $(s, a, s') \sim \mathcal{D}$ **do**
23:             $Q_{i,\xi}(s, a) \leftarrow Q_{i,\xi}(s, a) + \eta\Big[r_{\text{int}}^{(i,\xi)}(s, s') + \gamma \max_{a'} Q_{i,\xi}(s', a') - Q_{i,\xi}(s, a)\Big]$
24:          **end for**
25:       **until** $Q_{i,\xi}$ converges
26:       Define intra-option policy: $\pi_{i,\xi}(s) = \arg\max_a Q_{i,\xi}(s, a)$
27:       Define termination: $\beta_{i,\xi}(s) = \begin{cases} 1, & \text{if } Q_{i,\xi}(s, a) < 0, \ \forall a \\ 1/N, & \text{otherwise} \end{cases}$
28:    **end for**
29: **end for**
30: **Return:** $2k$ VPS options $\mathcal{O} = \{(\mathcal{S}, \pi_{i,\xi}, \beta_{i,\xi}) \mid i = 1, \ldots, k; \ \xi = \pm 1\}$

---

# C. Experimental Details and Additional Results

## C.1. Detailed Experimental Settings

All tabular experiments (GridWorld-Rooms, Taxi-v3) in this paper were conducted on an Intel i7-13700HX CPU, while all high-dimensional or continuous experiments (Visual GridWorld-Corridor, Atari 2600 games) were performed on a single Nvidia RTX 4090 GPU. The detailed experimental settings for each environment are as follows.

**GridWorld-Rooms**

- Environmental Settings

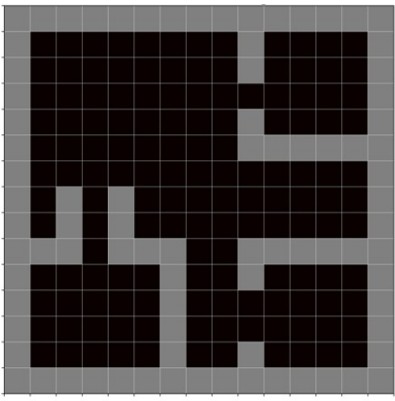

Figure 9: GridWorld-Rooms

Observation Space: 141 discrete states (corresponding to 141 free locations)
Action Space: 4 discrete actions (up, down, left, right)

- Option Training phase

| Parameter | VPS Option | Eigenoption | Random Option |
|---|---|---|---|
| Intrinsic reward number ($k_{\text{base}}$) | 10 | 10 | 10 |
| Option number | 20 | 20 | 20 |
| Discount for state-value function ($\gamma_V$) | 0.99 | – | – |
| Discount for Q-learning ($\gamma_Q$) | 0.99 | 0.99 | 0.99 |
| Learning rate for state-value function ($\alpha_V$) | 0.05 | – | – |
| Eligibility trace for TD($\lambda$) of $V$ ($\lambda$) | 0.9 | – | – |
| Learning rate for Q-learning ($\alpha_Q$) | 0.05 | 0.05 | 0.05 |
| Buffer episodes ($N_{\text{ep}}$) | 1000 | 1000 | 1000 |
| Episode length ($T_{\text{ep}}$) | 200 | 200 | 200 |

Table 2: Hyperparameter settings for option training in GridWorld-Rooms.

- Exploration Experiments

| Parameter | Value |
|---|---|
| Number of option sets ($n_{\text{outer}}$) | 10 |
| Runs per option set (inner) | 10 |
| Steps per run ($\max\_\text{steps}$) | 3000 |
| Expected option length $L$ | 15 |

Table 3: Exploration experiment core parameters in GridWorld-Rooms.

- Reward Collection Experiments

| Parameter | Value |
|---|---|
| Episodes per run | 100 |
| Steps per episode | 200 |
| Epsilon-greedy ($\epsilon$) | 0.1 |
| Q-learning learning rate ($\alpha$) | 0.05 |
| Discount factor ($\gamma$) | 0.99 |
| Option horizon $L$ | 15 |
| Runs per option set ($n_{\text{inner}}$) | 10 |
| Number of option sets ($n_{\text{outer}}$) | 10 |
| Reward for reaching goal | 1.0 |
| Step penalty | $-0.01$ |

Table 4: Core parameters for the reward collection experiment in GridWorld-Rooms. Each run is evaluated on the designated start-goal pair.

**Taxi-v3**

- Environmental Settings

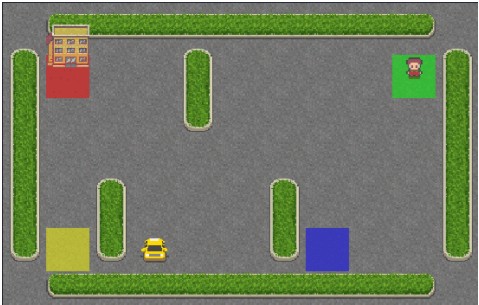

Figure 10: Taxi-v3 Environment

Observation Space: 500 discrete states (Taxi position, passenger location, destination)

Action Space: 6 discrete actions (south, north, east, west, pickup, dropoff)

Reward for successful dropoff: 20

Step penalty: $-1$

Invalid pickup/dropoff penalty: $-10$

- Option Training Phase

| Parameter | VPS Option | Eigenoption | Random Option |
|---|---|---|---|
| Intrinsic reward number ($k_{\text{base}}$) | 10 | 10 | 10 |
| Option number | 20 | 20 | 20 |
| Discount for state-value function ($\gamma_V$) | 0.999 | – | – |
| Discount for Q-learning ($\gamma_Q$) | 0.999 | 0.999 | 0.999 |
| Eligibility trace for TD($\lambda$) of $V$ ($\lambda$) | 0.9 | – | – |
| Learning rate for all Q-learning ($\alpha$) | 0.05 | 0.05 | 0.05 |
| Buffer episodes ($N_{\text{ep}}$) | 1000 | 1000 | 1000 |
| Episode length ($T_{\text{ep}}$) | 200 | 200 | 200 |
| Q-learning update steps ($N_{\text{step}}$) | 1,000,000 | 1,000,000 | 1,000,000 |

Table 5: Hyperparameter settings for option training in Taxi-v3.

- Random-Walk Task Completion Experiment

| Parameter | Value |
|---|---|
| Number of option sets ($n_{\text{outer}}$) | 10 |
| Episodes per set | 1000 |
| Max steps per episode | 200 |
| Option horizon ($L$) | 10 |
| Success criterion | Pickup and dropoff completed within an episode |
| Action selection | Uniform over primitive actions and options |

Table 6: Core parameters for the random walk task completion experiment in Taxi-v3.

- Reward Collection Experiments

| Parameter | Value |
|---|---|
| Episodes per run | 5000 |
| Steps per episode | 200 |
| Evaluation interval (episodes) | 10 |
| Evaluation trials per interval | 10 |
| $\epsilon$-greedy ($\epsilon$) | 0.1 |
| Q-learning learning rate ($\alpha$) | 0.1 |
| Discount factor ($\gamma$) | 0.95 |
| Option horizon ($L$) | 10 |
| Runs per option set ($n_{\text{inner}}$) | 10 |
| Number of option sets ($n_{\text{outer}}$) | 10 |

Table 7: Core parameters for the reward collection experiment in Taxi-v3.

**Visual GridWorld-Corridor and Atari 2600 Games**

- Environmental Settings

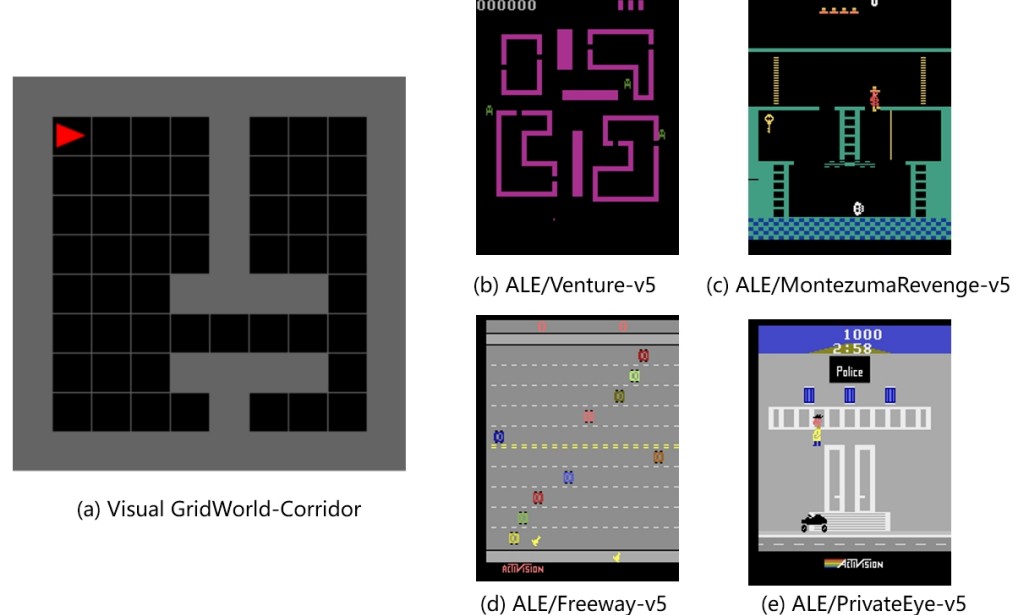

(a) Visual GridWorld-Corridor

(b) ALE/Venture-v5

(c) ALE/MontezumaRevenge-v5

(d) ALE/Freeway-v5

(e) ALE/PrivateEye-v5

Figure 11: Visual GridWorld-Corridor and Atari 2600 Environments

Observation Space: 84x84 grayscale images

- Network Structure

The neural networks employed for estimating VPS and the intra-policy of options in both the Visual GridWorld-Corridor environment and the Atari 2600 games share an identical CNN feature extraction backbone, as illustrated in Figure 12. The task-specific heads for each network are depicted in Figures 13 and 14, respectively. Here, $S_t$ denotes a preprocessed, single-frame grayscale image of size $84 \times 84$.

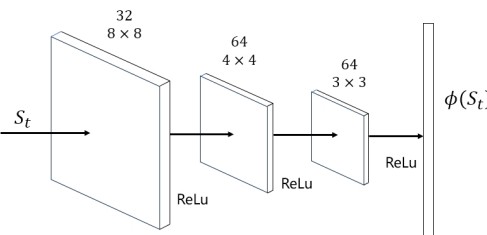

Figure 12: CNN for Visual GridWorld and Atari 2600 games

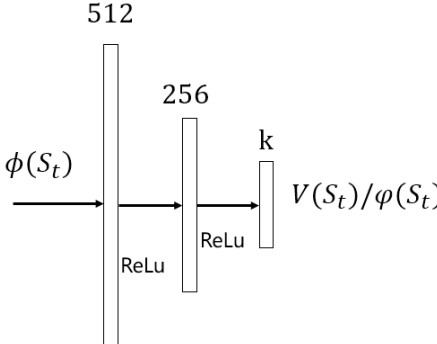

Figure 13: Value/VPS head for Visual GridWorld and Atari 2600 games

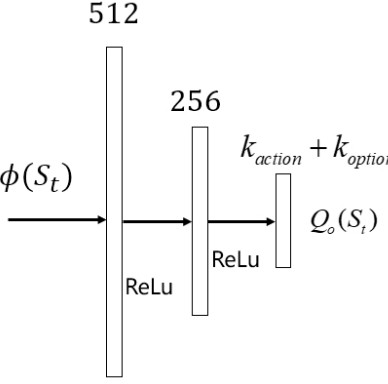

Figure 14: Option intra-policy head for Visual GridWorld and Atari 2600 games

- Hyperparameter Settings

| Parameter | Value |
| --- | --- |
| Number of options ($k_{\mathrm{opt}}$) | 8 |
| Discount for VPS ($\gamma_V$) | 0.99 |
| Discount for option Q-learning ($\gamma_Q$) | 0.9 |
| Buffer size | 500,000 |
| Value training iterations | 500,000 |
| VPS training iterations | 500,000 |
| Option-DQN training iterations | 500,000 |
| Batch size | 128 |
| Maximum episode length | 500 |
| Value network learning rate | $1 \times 10^{-3}$ |
| VPS network learning rate | $1 \times 10^{-4}$ |
| Option Q-network learning rate | $1 \times 10^{-4}$ |
| Value network loss function | Smooth L1 loss |
| VPS network loss function | Smooth L1 loss |
| Option Q-network loss function | MSE loss |

Table 8: Hyperparameter settings for VPS-based training in Visual GridWorld and Atari 2600.

## C.2. Supplementary Experimental Results

- VPS Distribution in Other GridWorld Environments

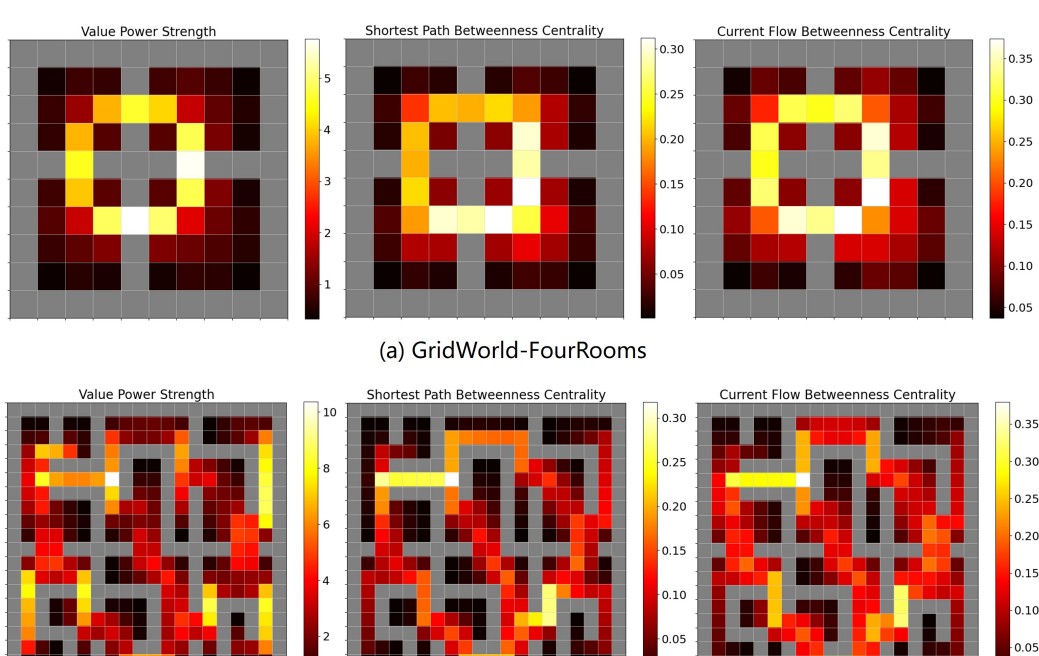

Figure 15: VPS Distribution in GridWorld-FourRooms and Maze

• A Group of 20 VPS Options in GridWorld-Rooms

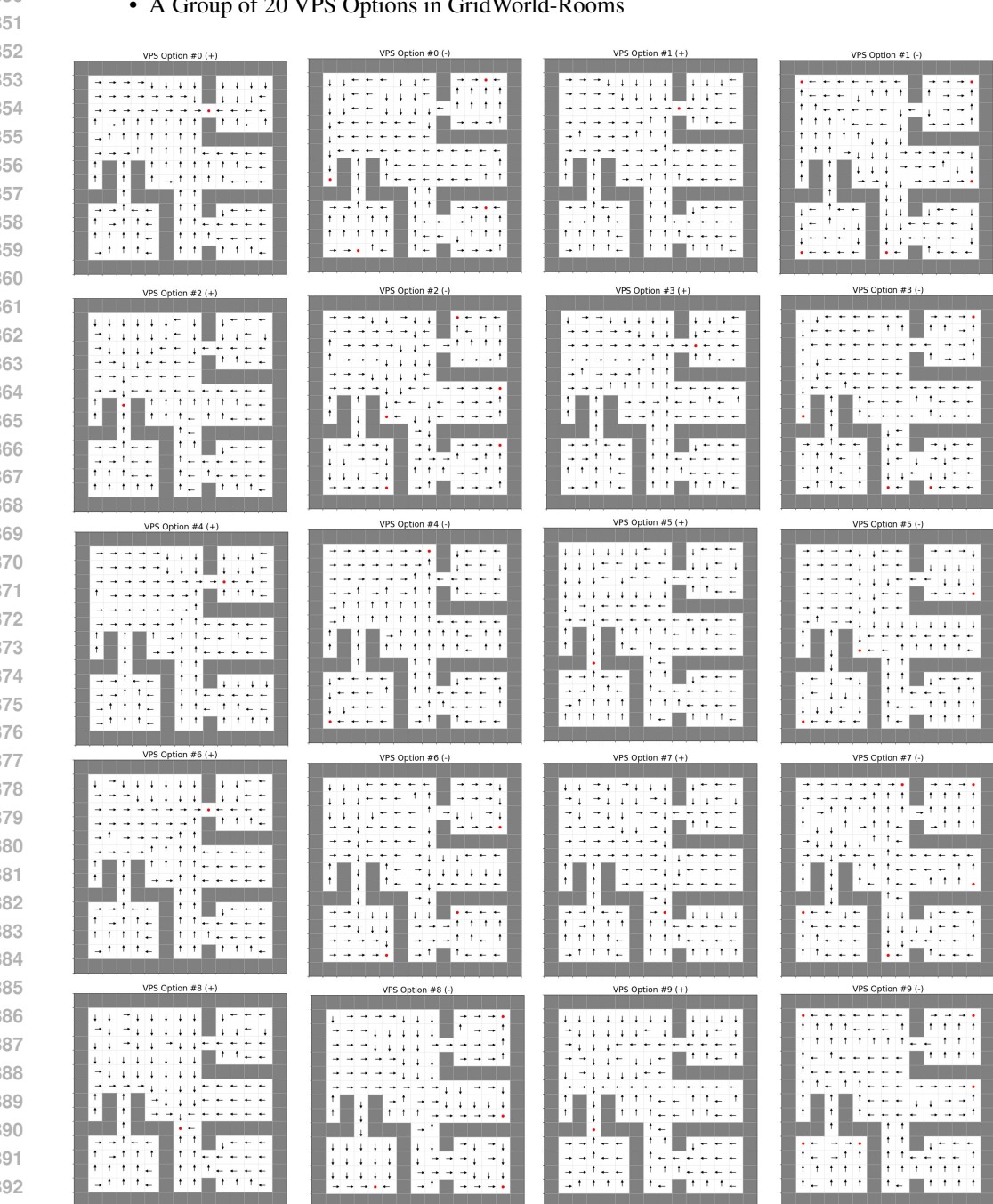

Figure 16: VPS Option Set #0 in GridWorld-Rooms

- VPS Peaks in Atari-PrivateEye, MontezumaRevenge and Freeway

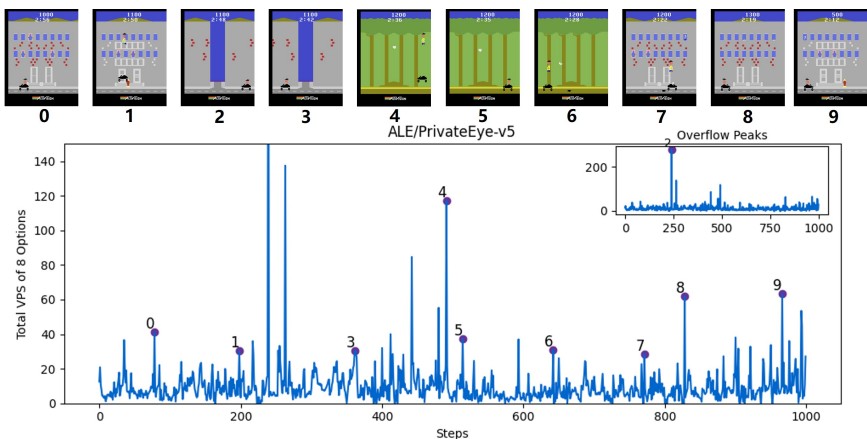

Figure 17: VPS Peaks in ALE/PrivateEye-v5

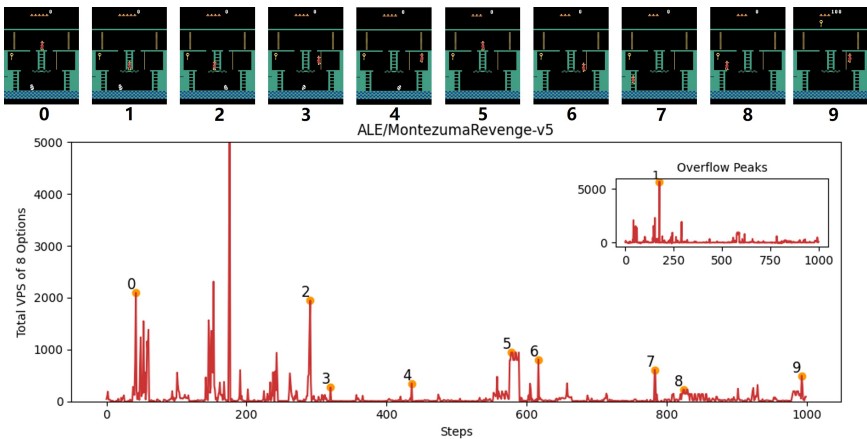

Figure 18: VPS Peaks in ALE/MontezumaRevenge-v5

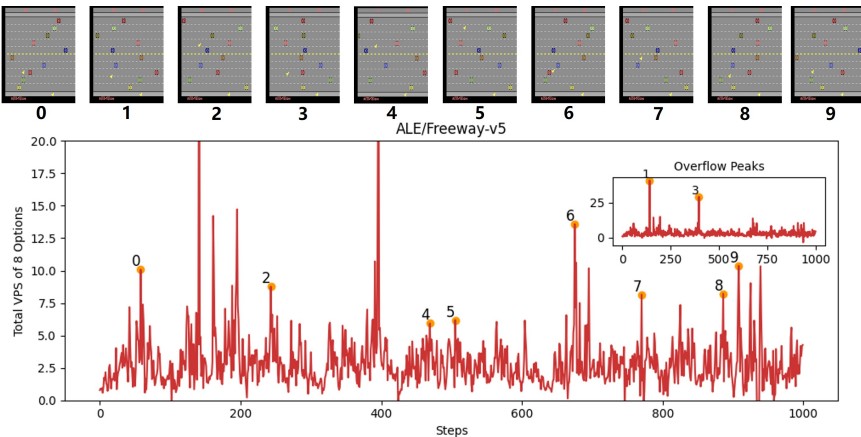

Figure 19: VPS Peaks in ALE/Freeway-v5

## D. Ethics Statement

This work adheres to the ICLR Code of Ethics. No human subjects, animal experiments, or personally identifiable information (PII) are involved. All environments are standard RL benchmarks or simulated grid worlds. We took care to avoid discriminatory outcomes; no demographic attributes are modeled or inferred. The method is evaluated only in research settings and is not deployed in safety-critical applications. We disclose the compute used and training budgets in the appendix to support transparency; we favor modest resources and standard hardware. We will promptly address any issues raised by the community regarding misuse or copyright/licensing concerns.

## E. Reproducibility Statement

We aim for full reproducibility. We will release an anonymized repository containing: (i) source code for value/VPS heads, option learning, and all baselines; (ii) configuration files with all hyper-parameters, random seeds, network architectures, optimizer settings, and training schedules; (iii) scripts to build environments, generate buffers, and reproduce every figure/table;

The paper includes implementation details (losses, targets, termination rules), evaluation protocols (number of seeds, episode lengths, and metrics), and ablations. With the provided instructions, a reader can reproduce the reported results on commodity GPUs/CPUs.

## F. LLM Usage

Large Language Models (LLMs) were used only for (a) language editing (clarity, grammar, and style) and (b) programming assistance (plotting scripts, minor refactoring, and debugging hints). LLMs did not contribute to research ideation, dataset creation, labeling, experiment design, result selection, or evaluation. All LLM-assisted code and text were reviewed and verified by the authors; the authors take full responsibility for the manuscript and for the final codebase. We ensured that any LLM-suggested snippets are compatible with our project's license and that no proprietary data or confidential information were provided to the model.

