# OpenReview forum: "Learning Interpretable Options by Identifying Reward Diffusion Bottlenecks in Reinforcement Learning"
_ICLR.cc/2026/Conference — Submitted to ICLR 2026_

### Official Review · Reviewer_BFjr · 2025-10-25

**Soundness:** 3
**Presentation:** 3
**Contribution:** 3
**Rating:** 6
**Confidence:** 3

**Summary:**

The paper introduces Value Power Strength (VPS), a value-based metric to detect bottlenecks in MDPs by exploiting a Bellman–Kirchhoff analogy. VPS is estimated online (with tabular convergence and a high-dimensional encoder extension) and used to discover options that steer agents toward/away from bottlenecks. Experiments on grid worlds and Atari show interpretable subgoals and stronger exploration than eigenoptions or random options.

**Strengths:**

- Originality: Introduces a value-based bottleneck metric (VPS) derived from a Bellman–Kirchhoff analogy—an elegant, non-graph alternative to spectral/eigen-option approaches.

- Quality: Provides tabular convergence guarantees for the VPS estimator under standard stochastic-approximation conditions.

- Clarity: Conceptual bridge from electrical networks to Bellman updates makes the VPS definition intuitive; figures/heatmaps clearly illustrate how bottlenecks emerge.

- Significance: Offers a practical path to interpretable option discovery that scales beyond tabular settings—useful for hierarchical RL and exploration in sparse-reward tasks.

**Weaknesses:**

- The paper frames VPS as a graph-free alternative to spectral/eigen-option methods, but the relation to prior “energy/Dirichlet” or Laplacian-based objectives is only qualitatively discussed. A tighter comparison (theory or controlled experiments) to Laplacian options/eigenoptions would clarify what is genuinely new versus a reparameterization of known smoothness/energy criteria.
Actionable: add a proposition showing when VPS ranks states similarly/differently from Laplacian energy or commute-time distances; include side-by-side heatmaps and correlation metrics of VPS vs. Laplacian scores on the same MDPs.

- VPS depends on value differences under a reference policy/data-collection regime. It is unclear how sensitive the bottleneck map is to (i) the random-reward distribution (Gaussian/RFF), (ii) the rollout policy (random walk vs. ε-greedy vs. learned), and (iii) the scale/shift of rewards.

-In function approximation, the value and VPS heads share an encoder; improvements could partly stem from representation sharing rather than the VPS objective per se.

-The option-construction rules (intrinsic reward based on VPS and termination around peaks) are heuristic. Their hyperparameters (peak detection thresholds, termination radius, number of options) seem influential.

**Questions:**

- Can you formalize when VPS will rank states similarly or differently from Laplacian-based scores (e.g., Dirichlet energy, eigenoptions)? A proposition relating $\phi_{\pi}(s)=E_{s'~P}[V_{\pi}(s)-V_{\pi}(s')]^2$ to cut conductance or effective resistance would clarify novelty beyond graph spectral smoothness. Please include side-by-side heatmaps and correlation analyses on the same MDPs.

- VPS depends on value differences under random-reward data and the behavior policy. How sensitive are VPS maps to (i) Gaussian vs. RFF random rewards, (ii) behavior policies (random-walk vs. $\varepsilon$-greedy vs. learned), and (iii) reward scaling? Please add rank-correlation and downstream option-quality curves across these choices.

- In visual settings, value and VPS share a backbone with separate heads. Could gains stem from representational sharing rather than the VPS objective? Add a capacity-matched control head (trained without the VPS loss) and compare option utility; report variance across seeds.

- Please compare to at least one skill-discovery method (e.g., DIAYN/APS) and one intrinsic-motivation method (e.g., RND/ICM) that impact exploration/coverage, matched on option budget/length.

---

### Official Review · Reviewer_JRsk · 2025-10-29

**Soundness:** 1
**Presentation:** 2
**Contribution:** 1
**Rating:** 2
**Confidence:** 5

**Summary:**

The authors proposed the Value Power Strength (VPS) framework for value-based option discovery. By learning the value function (through successor-representation decomposition) given random Gaussian rewards, the authors construct options based on the eigenoption framework (Machado et al., ICLR, 2018). The authors then proposed an extension of the framework to continuous state space, leveraging random Fourier features. The proposed method is evaluated on grid worlds (both discrete representation and continuous variant given visual inputs) and Atari benchmarks.

**Strengths:**

- The drawn analogy between the current flow network and RL is interesting and novel.
- The VPS framework indeed yields interpretable bottlenecks (but see below).

**Weaknesses:**

- The decomposition of value function in equations (2) and (3) are precisely the successor representation (SR) formulation. The authors should should cite relevant references (e.g., Dayan, Neural Computation, 1993; Stachenfeld et al., Nature Neuroscience, 2017), and discuss connections with the relevant RL algorithms that identify bottleneck states using SR-based metrics (Machado et al., JMLR, 2023; Yu et al., NeurIPS, 2023).
- In the drawn analogy between value function and the node voltage (or more broadly, between value iteration and current transmission), the key difference lies in the graph structure of the two problem. The power transmission assumes undirected graph whereas the transition operator in an MDP is cleary directed. This undermines the motivation behind the analogy between equation (7) and equation (11). There is no associated discussion on why the local Dirichlet energy in an undirected graph should yield equivalent interpretation under directed settings.
- Theorem 1 is the convergence proof of value iteration (or temporal difference learning), which is a well-known theorem proposed over three decades ago in Dayan, Machine Learning, 1992. The re-statement of the theorem in the main paper is redundant, and combined with the missing citations, collectively undermine both the soundness and contribution of the paper.
- The description of implementation details of figure 2 is crucial for understanding VPS, but is presented with great vagueness. Notably, the authors claim “placing a reward near a bottleneck effectively highlights it via VPS”, which indicates the necessity of placing rewards at locations of “importance” given prior information, which is contrary to the authors’ claim throughout the paper.
- Lemma 1 and Theorem 2 are re-statements of well-known results in linear algebra, and the application of such spectral decomposition has been exploited in previous works (Baram et al., BioRxiv, 2018; Yu et al., ICLR, 2021). This undermines the novelty of the proposed method, and the missing of relevant citations undermines the soundness of the presentation.
- The authors leverage eigenoptions for option-discovery given the value fucntions (given random rewards). I fail to see a clear improvement over the original eigenoption framework (Machado et al., ICLR, 2018). The “value function” computed in Theorem 2 (equation 15) bears high resemblance to the standard SR metric used for the original eigenoption framework. This significantly undermines the novelty of the proposed method.
- With the generalisation to the continuous domain with random Fourier features, the value function is no longer computed under the SR-like formulation, which violates the authors’ original proposal under the discrete case.
- As expected from the high resemblance of the proposed method and the SR-based eigenoption method, the performance is highly similar, both in terms of exploration efficiency and in terms of sample efficiency of RL. The gap in the case of (S1, G1), without further diagnostics, is more likely due to some stochastic initialisation error and some implementation differences (I did not check in the accompanying codebase).
- Figure 6b and 6c seems to yield difference interpretations. Figure 6b clearly suggests the eigenoption performs suboptimally comparing to the VPS framework, the but the training cue again show high similarity. There is not enough information to interpret what leads to the difference (and I strongly suggest the authors to expand the figure captions to be at least self-contained).

Minor points.
- The resolvent form (T = I - \gamma P_\pi) is neither the Bellman expected operator nor the Bellman optimality operator. The authors introduced $\mathcal{T}$ as a Bellman operator, which will be misleading to the readers.
- The weight function definition is confusing, it should represent the marginalised (over action) state transition probabilities, P(s’|s), rather than the action-dependent transition probability, P(s’|s, a).
- Bottom panel is figure 8 is stretched and blurred (should use svg to preserve image quality).

Typo.
- line 116: "an" -> "a".

References.

[1] Dayan, P., 1993. Improving generalization for temporal difference learning: The successor representation. Neural computation, 5(4), pp.613-624.

[2] Stachenfeld, K.L., Botvinick, M.M. and Gershman, S.J., 2017. The hippocampus as a predictive map. Nature neuroscience, 20(11), pp.1643-1653.

[3] Machado, M.C., Barreto, A., Precup, D. and Bowling, M., 2023. Temporal abstraction in reinforcement learning with the successor representation. Journal of machine learning research, 24(80), pp.1-69.

[4] Yu, C., Burgess, N., Sahani, M. and Gershman, S.J., 2023. Successor-predecessor intrinsic exploration. Advances in neural information processing systems, 36, pp.73021-73038.

[5] Dayan, P., 1992. The convergence of TD (λ) for general λ. Machine learning, 8(3), pp.341-362.

[6] Baram, A.B., Muller, T.H., Whittington, J.C. and Behrens, T.E., 2018. Intuitive planning: global navigation through cognitive maps based on grid-like codes. BioRxiv, p.421461.

[7] Yu, C., Behrens, T.E. and Burgess, N., 2020. Prediction and generalisation over directed actions by grid cells. arXiv preprint arXiv:2006.03355.

[8] Machado, M.C., Rosenbaum, C., Guo, X., Liu, M., Tesauro, G. and Campbell, M., 2017. Eigenoption discovery through the deep successor representation. arXiv preprint arXiv:1710.11089.

**Questions:**

See above.

---

### Official Review · Reviewer_SwBa · 2025-11-01

**Soundness:** 2
**Presentation:** 3
**Contribution:** 2
**Rating:** 4
**Confidence:** 5

**Summary:**

This paper introduces the concept of value power strength (VPS) as a proposal for scalable identification of bottleneck states in deep reinforcement learning. This contribution consists of three parts: the definition and motivation of VPS, a proposal for intrinsic reward functions from the VPS, and the empirical evaluation of a hierarchical reinforcement learning algorithm that makes use of options that maximize the proposed intrinsic rewards.

Regarding the definition, given a policy and a reward function, the VPS of a state is the expected squared value difference between the state and its successors. To motivate this definition, the paper identifies an equivalence relationship between the state graph determined by the transition probabilities and an electrical circuit where nodes are states and transition edges are resistors. Then, the squared value difference can be interpreted as energy dissipated in resistors and the bottlenecks as states where most of the current, or reward, flows.

From the circuit analogy, it is natural to define the VPS as an intrinsic reward function: a state with maximal intrinsic reward will correspond to a state with maximal inflow of reward. The problem, however, is that this definition is policy and reward dependent. The paper motivates the use of a uniform random policy based on a minimal empirical example. With respect to the choice of reward function, the paper proposes to use random reward vectors for the tabular case, where each entry is independent of the others, and random Fourier features for the continuous setting. To explain the choice, the paper argues that random vectors will lead to value functions that will lie mostly on the top eigenspace of the graph Laplacian, i.e., they will be composed of low frequency components.

Finally, the paper provides four empirical results to assess the effectiveness of the proposed approach to automatically find sets of options with bottleneck-seeking behavior. In particular, the experiments evaluate the state coverage, return curves, and task completions in tabular environments, and the scalability of bottleneck identification to high-dimensional visual inputs.

**Strengths:**

This work presents an interesting and, to the best of my knowledge, novel take on the state-transition graph. By identifying edges with resistors and inverse transition probabilities with resistances, the paper clearly motivates the use of the equivalent of electric power to identify bottlenecks. This identification is interesting in itself and might lead to further connections with electrical engineering or physics.

Further, the paper is conceptually clear regarding the background concepts, the introduced definition, and the algorithm proposed for option discovery.

**Weaknesses:**

**Major weaknesses**

I identify two main groups of weaknesses: the motivation for the proposed bottleneck-identification and option-discovery algorithm is not satisfactory and the lack of details in the experimental section makes it hard to assess the generality or validity of the conclusions.

*Motivation*: The proposed algorithm is deeply connected to the concept of eigenoptions but it fails to clearly elucidate the connection and to motivate the need for the proposed method in light of the existence of eigenoptions.

First of all, the paper seems to suggest that there are no previous methods to identify and reach bottlenecks in continuous settings. Nevertheless, there exists a large number of works dealing with the problem of learning the top eigenspace of the graph Laplacian [1-4], which in turn means that eigenoption discovery is completely scalable to the continuous setting. These options have been shown to naturally seek bottlenecks.

Second, the motivation to use random reward functions for the choice of VPS is that their value functions will correspond to lower frequency components. The problem with this motivation is that the reward functions with lowest frequency components are precisely the bottom eigenvectors of the graph Laplacian, which are precisely the intrinsic reward functions used to define the eigenoptions. While this is not equivalent to choosing the eigenvectors to define the VPS and then finding options that optimize their VPSs, it is highly likely that the nodes with higher power will be the same as those with higher entries in the eigenvectors (because of the smoothness of the eigenvectors). Hence, it is to be expected that the eigenoptions will behave very similarly to those options optimizing the "eigen-VPSs". This raises the question about the need or advantage of sampling reward functions and learning their corresponding value functions if there are already methods to identify eigenvectors, which are already value functions by definition (which means that there is no need to learn their values).

Third, for the continuous state space setting, the paper proposes to use random Fourier features (RFFs) in the space of some learned representation of the states. There are two problems with this choice: 1) the rewards are not stationary since they depend on the representations learned in the process of estimating the value functions. It is unclear to me whether there exists any equilibrium point for such dynamics, which are not covered by the mathematical analysis in Theorem 1. 2) The rationale for choosing RFFs is that intrinsic rewards should be smooth on the state-transition graph. However, again, this is a property that is already satisfied by the eigenvectors of the graph Laplacian. This begs the question of why one would prefer to use a possibly unstable alternative that has the same desirable feature.

*Missing experimental details*:

- Above all, the paper does not explain the hierarchical reinforcement learning used. It is clear how options are learned, but not how they are used to learn value functions. Considering this, it is hard to make any comparison between learning curves.
- It is not stated what reward function or algorithm is driving the state coverage in the exploratory experiments. Hence, it is unclear whether there is any meaningful difference between the option discovery methods.
- Figure 5 shows learning curves for 3 particular states and goals, but the average results are not included. Again, this makes it harder to conclude anything about the difference in learning performances.
- It is unclear how Figure 6b) was obtained. In particular, there is no explanation for what task completion with random walk means or why this metric is relevant.
- There is no explanation for what the different symbols mean in Figure 8. In particular, it is unclear why some frames have stars and not others, what the metric "total VPS of options" is,  and what criterion is used to choose numbered frames, given that there are many peaks without numbers that are higher than some of the numbered ones. Most importantly, it is unclear what percentage of peaks actually correspond to bottleneck states and so it is hard to conclude anything about the reliability of "total VPS" to identify bottlenecks in the high-dimensional setting from the provided evidence.

**Minor weaknesses**
- The number of seeds in the tabular case is too small to be able to tell any significant difference. Given that the environments are small, computational costs should not be a problem. In addition, given that the similarity with the eigenoption approach is considerable, it seems important to have precise experiments (i.e., with a lot of samples) that capture small differences.

**References**
- [1] Yifan Wu, George Tucker, and Ofir Nachum. The Laplacian in RL: Learning Representations with Efficient Approximations. In International Conference on Learning Representations (ICLR), 2019.
- [2] Diego Gomez, Michael Bowling, and Marlos C. Machado. Proper Laplacian Representation Learning. In International Conference on Learning Representations (ICLR), 2024.
- [3] Ahmed Touati, Jérémy Rapin, and Yann Ollivier. Does Zero-Shot Reinforcement Learning Exist? In International Conference on Learning Representations (ICLR), 2023.
- [4] J. Jon Ryu, Samuel Zhou, and Gregory W. Wornell. Revisiting Orbital Minimization Method for Neural Operator Decomposition. Advances in Neural Information Processing Systems (NeurIPS), 2025.

**Questions:**

**Suggestions**

In order of relevance:
- Additional motivations or experiments should be included to explain differences or advantages with respect to eigenoptions.
- Add missing experimental details.
- Correct comments that suggest that the scalability to high-dimensional or continuous domains of state-transition graph methods is limited.
- I would place the theorems in the appendix. In my opinion, they are standard results and not really contributions. However, they are relevant for the discussion and so I think it makes sense to include them in the paper. This will give some room to further add experimental details that might have been obviated in the light of missing space.
- In the spirit of the previous suggestion, I would not name theorems as such, but just lemmas.
- The Bellman operator is not defined as stated in the first line of page 3 (at least not usually). Given that the operator is never used in any other part, I would refrain from introducing it, much less with such an atypical definition.

**Questions**
- It is unclear what "random seeds" means in the context of the evaluation process. Does this mean that start and goal states are chosen uniformly?
- Why do you use a shared encoder?

---

### Official Review · Reviewer_2ZNE · 2025-11-01

**Soundness:** 3
**Presentation:** 2
**Contribution:** 2
**Rating:** 4
**Confidence:** 3

**Summary:**

The paper presents a new method for discovering options, in the sense of temporally-extended high-level actions, in reinforcement learning environments. The underlying method is based on an analogy between the Bellman equation and Kirchoff's electrical current law. This connection is used to define a quantity called Value Power Strength. Options are then defined to have policies which maximize expected return under an intrinsic reward based on the VPS associated with random rewards. The method is extended to continuous environments using random fourier features. The authors present results on the ability of the learned options to cover states in the environment, and to accelerate learning, in classical tabular environments. Also, the qualitative attributes of the learned options in environments from the Atari Learning Environment are shown.

**Strengths:**

Originality: The connection between Kirchhoff's current law and the Bellman equation is interesting and novel (to my knowledge).

Quality: The mathematical framework appears well-thought-out / sound.

Clarity: The notation is mostly sharp / clear, and gridworld heatmap figures / Atari visualizations demonstrate that VPS finds bottleneck states.

Significance: The design of effective options is an open and important research problem, and I appreciate that this work offers a different perspective in this direction.

**Weaknesses:**

- The paper lacks context / motivation for the principles of electricity that are used. I believe the paper would be strengthened by answering these questions for the reader in the beginning: What is Kirchhoff's current law? What does studying options through the lens of Kirchhoff's current law give us, beyond just using the Bellman equation?
- I'm not sure that the paper is properly contextualized within the related work. For example, the authors say, referring to bottleneck identification methods, that "these graph-centric pipelines do not scale gracefully to high-dimensional or continuous state spaces". However, Klissarov & Machado 2023 present results on such tasks. Can the authors comment on how their method differs from or is similar to Laplacian-based methods such as this one?
- Relatedly, can the authors address how their random reward scheme is related to other methods which build on the value functions associated with random cumulants, such as Farebrother et al. 2023, "Proto-Value Networks"?
- The writing quality could be improved overall, and the figure captions are quite sparse, which can make it hard to understand the figures / results themselves. For example, the experiment description on line 251 lacks enough detail to understand exactly what exactly is being visualized, e.g. I can't quite parse what "sequentially assigning a reward of 1 to each grid and aggregating the resulting VPS distributions" means.
- While I am reasonably convinced that the proposed method VPS is able to identify bottleneck states in the environment, the figures in Section 4 don't suggest that VPS is much better than the baseline for state coverage, and it doesn't appear to improve at all over random/eigen options for learning. Ultimately, this leaves me feeling skeptical of the potential for downstream impact of this work.

Nits:
- I believe that Equations are first-class objects along with Tables, Figures, etc – "equation 2" should be "Equation 2"
- "According to Table 1, while the node current flow equation 6 depicts the bottlenecks of charge flow, VPS essentially depicts the bottlenecks of reward diffusion in the state space." This sentence is very confusing to me as a reader because the combined information present in Table 1 and Equation 6 does not obviously suggest to me that VPS "depicts the bottlenecks of reward diffusion in the state space". Perhaps the authors could explain more directly, rather than relying on references.
- "Since the value of VPS depends on the value function Vπ , a simultaneous update law of Vπ and φπ has been proposed as below." Who has proposed it? Is this a new contribution of this work? I think more active language would benefit the reader here – i.e. "we propose a simultaneous update law".

**Questions:**

- Can the authors provide any intuition / motivation for why the VPS equation (Equation 11) ends up representing bottleneck states? I.e. why does the squared difference in value between neighboring states model this?
- Perhaps related, out of curiosity, are there "n-step" generalizations of VPS? How would you expect them to behave differently?
- I don't fully understand the termination condition for the options. Why is it reasonable to check for cases where the optimal value function under the intrinsic reward is less than 0? What is this doing?

---

> ### Author Response · Authors · 2025-11-28
> **Response to the comments of Reviewer 2ZNE**
>
> We sincerely thank the reviewer for the thoughtful and constructive comments. These suggestions significantly improved the clarity, motivation, and positioning of our work. Below we restate each comment  and provide our responses. All changes appear in red in the revised manuscript.
>
> **Weakness**
>
> **W1. “The paper lacks context/motivation for the principles of electricity… What is Kirchhoff’s current law? What additional value does this analogy provide beyond the Bellman equation?”**
>
> Thank you for pointing this out. In the revised Introduction (Page 2), we added a new paragraph that introduces Kirchhoff’s current law (KCL) and explains why drawing an analogy between electrical circuits and the Bellman equation provides a novel perspective for option discovery. Specifically, we clarify that circuit theory contains physically meaningful quantities—such as node power—that play central roles in analyzing flow bottlenecks in electrical networks, yet have received little attention in RL. By mapping these quantities to their Bellman-equation counterparts, this analogy highlights alternative structural information in MDPs that is not captured by standard value-based analysis alone
>
> **W2. “I'm not sure that the paper is properly contextualized within the related work...”**
>
> We thank the reviewer for this observation. The previous version indeed lacked sufficient context. We added a dedicated Related Work section (Page 9), where we compare VPS with proto-value functions, eigenoptions, Laplacian-based spectral methods, and deep Laplacian representation learning. We highlight that while Laplacian eigenvectors capture global smoothness directions of the state-transition graph, VPS measures local Dirichlet energy of the learned value function and therefore directly identifies bottleneck states rather than general smoothness components.
>
> **W3. “Relatedly, can the authors address how their random reward scheme is related to other methods which build on the value functions associated with random cumulants, such as Farebrother et al. 2023, `Proto-Value Networks'?”**
>
> Thank you for your comments. In the last paragraph of the Related Work section (Page 10), we haved compared these methods with VPS option framework.
>
> **W4. “The writing quality could be improved overall, and the figure captions are quite sparse...”**
>
> We apologize for this lack of clarity. The caption of Figure 2 has been fully rewritten (Page 4) to clarify what the results mean.
>
> **W5. “While I am reasonably convinced that the proposed method VPS is able to identify bottleneck states in the environment, the figures in Section 4 don't suggest that VPS is much better than the baseline for state coverage, and it doesn't appear to improve at all over random/eigen options for learning. Ultimately, this leaves me feeling skeptical of the potential for downstream impact of this work.”**
>
> Thank you for the comment. Different option–discovery methods suit different scenarios. Laplacian-based methods such as eigenoptions aim to span smooth directions of the state-transition graph and, as more options are added, they naturally distribute exploration across the whole space, reducing wandering and improving state coverage.
>
> In contrast, VPS options focus purely on bottleneck states. If a GridWorld contains only a few ``door” states, then any number of VPS options will concentrate on directions toward (or away from) these bottlenecks. Thus, when bottlenecks are essential for completing the task, VPS options can be advantageous; but if the metric is state coverage alone, eigenoptions will surpass VPS once enough of them are provided (However, when the number of options are limited to a relevent small range, VPS option is found to perform better than eigenoption for state coverage).
>
> VPS method naturally suits tasks where bottleneck structure is crucial. In our current results, Taxi-v3 (Figure 6) is one such example, since picking up/dropping off passengers requires transitioning through bottleneck states. We believe VPS will also be helpful in more complex environments where bottlenecks dominate exploration difficulty.
>
> **W6. I believe that Equations are first-class objects along with Tables, Figures, etc – "equation 2" should be "Equation 2"**
>
> Thank you for the reminder. We have addressed this issue and double checked similar typo errors in the revised manuscript.
>
> **W7. ``This sentence is very confusing to me as a reader ...''**
>
> We apologize for the confusion—this was partially caused by a reference error. The correct reference should have been Equation 7, not Equation 6. In the revised manuscript we corrected this and added a clearer explanation in Page 5.
>
> **W8. “Who has proposed it? Is this a new contribution of this work?...”**
>
> We apologize for the unclear expression. The text has been updated on Page 5 to explicitly state that ``we have proposed a simultaneous update law''.

---

> ### Author Response · Authors · 2025-11-28
> **Response to the questions of Reviewer 2ZNE**
>
> Thank you very much for your questions which are interesting and instructive to us. Our response is as follows, representing our personal views for communication with you.
>
> **Q1. “Can the authors provide any intuition / motivation for why the VPS equation (Equation 11) ends up representing bottleneck states? I.e. why does the squared difference in value between neighboring states model this?”**
>
> Thank you for the question. The motivation for introducing VPS comes from circuit theory: node electrical power is used to quantify how much power dissipation is routed through each node, i.e., how responsible that node is for the overall energy dissipation pattern in a resistive network. For a more direct intuition in our setting, we can view this through a graph-theoretic lens. Dirichlet energy characterizes the smoothness of a graph signal, and Equation 8 shows that VPS is exactly the node-wise component of this energy. States with high VPS are therefore those where the graph signal changes sharply across edges, which corresponds to classical bottleneck (or cut) regions in graph theory (In graph theory, it is generally believed that the parts of the graph where the signal changes significantly are more likely to be the bottlenecks within the graph).
>
> **Q2. “Perhaps related, out of curiosity, are there ‘n-step’ generalizations of VPS? How would you expect them to behave differently?”**
>
> This is very close to a direction we are now considering. The current VPS update (Equation 13) is a one-step simultaneous update of the value function and VPS. A known limitation is that VPS lags behind value estimation and becomes accurate only after the value function itself has nearly converged. We have explored whether an approximate or n-step formulation could compute VPS more efficiently—for example, reformulating VPS directly as a value function and estimating it with the efficient $$TD(\lambda)$$ method. So far, we have not identified a principled solution. We are glad you raised this question, as it encourages us to continue investigating more effective formulations of VPS.
>
> **Q3. “I don't fully understand the termination condition for the options. Why is it reasonable to check for cases where the optimal value function under the intrinsic reward is less than 0? What is this doing?”**
>
> Thank you for the question. Our option termination rule follows the standard potential-based construction as in eigenoptions. We explain the mechanism more explicitly below.
>
> We define the intrinsic reward as
> $$ r_{int} = \phi(s') - \phi(s), $$
> where $$\phi(s)$$ is a potential function over states. Under this intrinsic reward, the optimal intrinsic value function $$ Q^\*_{int}(s,a)$$
> encourages the agent to move in directions that increase the potential.
>
> Thus, the greedy policy with respect to $$Q^\*_{int}$$ always make the agent climb the potential peak toward
> $$ s^\* = \arg\max_s \phi(s). $$
>
> Once the agent reaches $$s^\*$$, the following hold:
>
> 1. The potential cannot increase any further, so for any successor $$s'$$ we have
>    $$ \phi(s') \le \phi(s^\*) . $$
>
> 2. Consequently, all intrinsic rewards at $$s^\*$$ are non-positive:
>    $$ r_{int}(s^\*, a) = \phi(s') - \phi(s^\*) \le 0. $$
>
> 3. Therefore, all intrinsic optimal action values satisfy
>    $$ Q^\*_{int}(s^\*, a) \le 0, \quad \forall a. $$
>
> Thus, checking when
> $$ Q^\*_{int}(s, a) < 0 $$
> is equivalent to detecting that the agent has already reached the region of maximal potential for this option. At this point, the option has effectively completed its intended behavior, since no further action can increase its potential.
>
> For this reason, the condition $$Q^\*_{int}(s,a) < 0$$ is the standard and natural termination criterion for potential-based options.

---

### Meta-Review · Area_Chair_JiY8 · 2025-12-22

**Summary:**

All reviewers raised concerns about the similarity with Laplacian option-discovery methods. Given that the authors did not respond to this major concern during the rebuttal period, I recommend that the paper be rejected.

My understanding of the reviews is _not_ that they're saying that the voltage/Kirchoff interpretation of skill learning is unoriginal, but rather that the resulting algorithm bears a strong resemblance to Laplacian-based skill learning methods. Thus, one potential route for the authors would be to reframe the paper not to be about a new method, but rather to be about a new interpretation of an old method. The crux, then, would be figuring out what this new interpretation buys us. E.g., does this suggest scenarios when Laplacian-based methods should fail/succeed? Does it suggest theoretical results that can be applied to Laplacian-based methods?

**Reviewer Concerns:**

(see below)

**Reviewer Scores:**

2ZNE: 4
* paper lacks context / motivation for the principles of electricity that are used
* similarity to Laplacian methods
* how their random reward scheme is related to other methods which build on the value functions associated with random cumulants
* writing quality could be improved overall
* figures in Section 4 don't suggest that VPS is much better than the baseline for state coverage

SwBa: 4
* The proposed algorithm is deeply connected to the concept of eigenoptions but it fails to clearly elucidate the connection and to motivate the need for the proposed method in light of the existence of eigenoptions.
* Missing experimental details

JRsk: 2
* relationship with the SR and eigenoptions
* unclear analogy between value function and node voltage
* proper attribution of theoretical results to prior work


BFjr: 6
* relation to prior “energy/Dirichlet” or Laplacian-based objectives is only qualitatively discussed
* sensitivity to reward distribution, rollout policy, scale/shift of rewards.

---

### Decision · Program_Chairs · 2026-01-26

Reject